# Ab Initio Modelling of the Structure of ToxA-like and MAX Fungal Effector Proteins

**DOI:** 10.3390/ijms24076262

**Published:** 2023-03-26

**Authors:** Lina Rozano, Yvonne M. Mukuka, James K. Hane, Ricardo L. Mancera

**Affiliations:** 1Curtin Medical School, Curtin Health Innovation Research Institute, GPO Box U1987, Perth, WA 6845, Australia; 2Curtin Institute for Computation, Curtin University, GPO Box U1987, Perth, WA 6845, Australia; 3Centre for Crop and Disease Management, School of Molecular and Life Sciences, Curtin University, GPO Box U1987, Perth, WA 6845, Australia

**Keywords:** fungal effector protein, ToxA-like effector family, MAX effector family, Rosetta, TM-score, MaxCluster, ab initio modelling

## Abstract

Pathogenic fungal diseases in crops are mediated by the release of effector proteins that facilitate infection. Characterising the structure of these fungal effectors is vital to understanding their virulence mechanisms and interactions with their hosts, which is crucial in the breeding of plant cultivars for disease resistance. Several effectors have been identified and validated experimentally; however, their lack of sequence conservation often impedes the identification and prediction of their structure using sequence similarity approaches. Structural similarity has, nonetheless, been observed within fungal effector protein families, creating interest in validating the use of computational methods to predict their tertiary structure from their sequence. We used Rosetta ab initio modelling to predict the structures of members of the ToxA-like and MAX effector families for which experimental structures are known to validate this method. An optimised approach was then used to predict the structures of phenotypically validated effectors lacking known structures. Rosetta was found to successfully predict the structure of fungal effectors in the ToxA-like and MAX families, as well as phenotypically validated but structurally unconfirmed effector sequences. Interestingly, potential new effector structural families were identified on the basis of comparisons with structural homologues and the identification of associated protein domains.

## 1. Introduction

Fungi are considered a major threat to global food security, and it is estimated that the gross production of important crops drops by 13% each year due to fungal infection (FAOSTAT, 2017). Many fungal pathogens release cytotoxic or virulence-promoting “effector” proteins that mediate disease infection in plants, including agriculturally important crops such as rice, wheat and maize [1]. An effective strategy to control disease infection in crops is by breeding crops that carry resistance (R) genes or lack sensitivity (S) genes specific to certain fungal effectors [2].

Secreted effector proteins may either reside in the plant cytoplasm (cytoplasmic effectors) or in the plant apoplastic space (apoplastic effectors) [3,4]. Effector proteins usually have less than 250 amino acid residues and contain a relatively high number of cysteines [5]. However, the discovery of new fungal effector proteins solely based on their sequence is challenging since there is little protein sequence conservation among most known effectors [6,7]. Sequence-based prediction tools such as EffectorP [8] and Predector [9] implement machine learning and have made incremental improvements in distinguishing effector proteins from non-effectors [10,11,12]. However, achieving reliable effector prediction remains a significant challenge in molecular plant pathology.

In recent years, a growing number of studies have highlighted striking similarities of three-dimensional (3D) protein structures between effector proteins that were otherwise dissimilar at the sequence level. These similarities were observed following X-ray diffraction (XRD) and nuclear magnetic resonance (NMR) protein structure determinations, leading to the identification of structural families [13,14,15]. The “ToxA-like” and “MAX” (Magnaporthe AVRs and ToxB-like) families are the two best-studied families containing the largest number of 3D structures available [14]. The ToxA structural family contains structures from *Pyrenophora tritici repentis*, *Fusarium oxysporum* f. sp. *lycopersici* and *Melampsora lini* [16], whilst MAX contains structures from *P. tritici repentis* and *Magnaporthe oryzae* [17,18]. Both ToxA-like and MAX families are characterized by the presence of a β-sandwich fold [17,19]. Other emerging families, including AvrP, AvrLm4-7, knottin-like, nuclear transport factor 2-like (NTF2), SnTox3, necrosis-like proteins (NLPs), and RALPHs currently have only a single structure available [13,20,21].

A common structural feature of the ToxA-like family is a β-sandwich fold formed by eight or seven β-strands. Currently, the 3D structures of only three fungal effector proteins that belong to the ToxA-like structural family have been determined: ToxA from *P. triitici repentis* (PDB ID: 1ZLD) [19], Avr2 from *Fusarium oxysporum* f. sp. *lycopersici* (*Fol*) (PDB ID: 5OD4) [22], and AvrL567 from *Melampsora lini* (PDB ID: 2QVT) [23]. Despite sharing a similar protein fold, each has unique structural features: (a) ToxA has a 3_10_ helix at its N-terminus [19], (b) Avr2 (*Fol*) contains a π-helix between β-strands 6 and 7, followed by a long disordered loop that connects β-strands 7 and 8 [22], and (c) AvrL567 contains a 3_10_ helix located in a long disordered loop connecting β-strands 6 and 7 (Figure 1) [23].

The MAX structural family includes ToxB from *P. tritici repentis*, and AvrPiz-t, Avr1CO39, AvrPia, AvrPib and AvrPik/APikL2 variants from *M. oryzae* [14] and the MAX family has the most abundant available experimental 3D structures compared to other effector structural families. MAX effector proteins share a common β-sandwich fold consisting of six β-strands and either one or two disulfide bridges (Figure 2), except for AvrPib, which has no disulfide bridge. The structure of ToxB, Avr1CO39, AvrPia and AvrPik variants consist of five long β-strands, with one short β-strand (strand 5) positioned in a long disordered loop connecting β-strand 4 and 6 [15,17,18,25,26,27,28]. AvrPiz-t also contains five long β-strands and has a long disordered C-terminal tail with a short β-strand (strand 5) [29]. AvrPib is a recently reported MAX effector which consists of four long β-strands and two short β-strands, with no disordered loop region within its structure [14]. The second common β-strand in the MAX structural family is known to be the site of interaction with plant binding-proteins containing a heavy metal-associated (HMA) domain [25,27].

Available 3D structures of confirmed fungal effector proteins could enable the prediction of novel fungal effector candidates based on structural similarity to known effector families using template-based modelling, which we have indeed successfully applied to the prediction of the structure of candidate effector sequences possessing structural similarity to ToXA- and MAX-like effectors [24]. However, the limited number of available fungal effector protein structures restricts the application of this approach. This poses a substantial challenge for extending the prediction of emerging effector families defined by common structure (e.g., the ToxA-like and MAX-like families) to numerous predicted effector families that do not yet have available experimentally confirmed structures [30]. In these cases, computational modelling is necessary for the structural prediction of structurally unconfirmed effectors and effector candidates, such as through the use of ab initio methods.

Ab initio modelling is the prediction of protein structures based on their sequences without the use of any template structure. The success of the ab initio approach is mainly due to its use of knowledge- and physics-based approaches during the modelling of the protein structure. Rosetta ab initio is one of the top-ranked tools in the Critical Assessment of Structure Prediction (CASP) [31] owing to its use of short fragments obtained from known folds [32]. A limitation of ab initio modelling is that it is a computationally expensive task, and the structure-scoring functions still require improvement. The increased accuracy of ab initio prediction can be obtained with the use of structural restraints, such as protein contact maps, disulfide bonds and evolutionary information, which also depends on the availability of a match with the target protein [33,34,35,36,37].

A prior study reported the application of ab initio modelling for the prediction of the structure of fungal effector proteins of *Magnaporthe oryzae* [38] and the use of deep-learning approaches in the modelling of fungal effectors [39]. Our study broadens the application of ab initio modelling to multiple fungal pathogen species to determine if it can successfully predict experimentally confirmed structures of effector proteins from the well-defined ToxA and MAX families. These two structural families were chosen not only because of their impact on economically important cereal crops but also because they have the largest number of experimentally determined structures available compared to other fungal effector families. We independently (without the assistance of a ToxA or MAX structural template) predicted the structures of the sequences of ToxA-like and MAX effector proteins with known structures to determine the best parameters for modelling. This method was then broadly applied to predict the structures of (a) putative and structurally unconfirmed members of the ToxA-like and MAX families that were solely predicted by sequence and HMM-based methods and (b) other structurally unconfirmed fungal effector proteins [11].

## 2. Results

### 2.1. Benchmarking Metrics for Assessing the Successful Prediction of Ab Initio Model Structures

To our knowledge, this is the first study reporting the use of Rosetta ab initio modelling of ToxA-like and MAX effector proteins. We evaluated the success of this approach using a range of numbers of model structures (*nstruct*) generated per run, from 1000 to 50,000. A value of 30,000 was determined to be the minimum *nstruct* required to generate the best ab initio model that was close to the template native structure (see the section below on the clustering of structural models of ToxA-like and MAX effectors). This *nstruct* value was then applied to the modelling of structurally unconfirmed effectors. The choice of the value of *nstruct* also reflected the computational cost involved, and further analysis and discussion of the use of *nstruct* = 30,000 is presented with the clustering data (based on data in Section 2.3).

For the selection of native-like models for ToxA-like and MAX effectors, the predicted ab initio models were superimposed with their corresponding native structures, and their structural similarities were assessed based on Cα RMSD and TM-score values. A comparison of the use of RMSD and TM-score values to identify the best models was conducted using plots of REU against RMSD and TM-score for each effector template, as shown in Figure 3 and Figure 4. REU is a common quantitative metric used by Rosetta to determine structural models that fold better than those with the most negative REU values. These plots reveal a funnel-shaped energy landscape for most of the effector proteins as a function of RMSD, which follows the traditional funnel-shaped landscape of protein folding, whereby the free energy decreases gradually towards the native conformation basin, where the most native-like conformations are to be found [40]. By contrast, an inverted funnel-shaped energy landscape is observed with TM-score values, which is due to the fact that higher TM-score values indicate better structural alignment, whereas low RMSD values indicate better structural alignment.

Initially, three independent criteria were considered to identify the best structural model predicted by Rosetta: (i) the most negative REU value, (ii) the lowest RMSD, and (iii) the highest TM-score. In order to select the best criteria, assessments were made based using REU vs. RMSD and REU vs. TM-score plots.

In all of the REU vs. RMSD plots (left-hand side panels in Figure 3 and Figure 4), a trend was observed where the most negative REU values did not correlate with low-RMSD values (RMSD values below 3.0 Å), which represent ab initio models with the closest similarity to the native structure. For example, in the case of Tox-A effectors, the model with the lowest REU value of −230.0 had an RMSD of 14.5 Å (Figure 3A), and the model with the lowest REU of −190 had an RMSD of 9.5 Å (Figure 3C). In Figure 3B, the REU vs. RMSD plot shifted more to the left with a slightly lower overall RMSD distribution; however, the lowest REU of −209 had an RMSD of 10.0 Å. This trend was the same for MAX effector templates (Figure 4), where the lowest REU values of −138 (Figure 4C), −134 (Figure 4F), and −110 (Figure 4E) had RMSD values of 16.2, 13.2 and 11.3 Å, respectively. Similarly, the highest REU values of −97 (Figure 4B), −89 (Figure 4D), and −87 (Figure 4A) had RMSD values of 5.2, 5.0, and 10.0 Å, respectively.

Similar to the RMSD comparisons above, the comparison of REU vs. TM-scores also indicated that the most negative REU values also did not correspond to the best TM-scores (the most positive) (right-hand side panels in Figure 3 and Figure 4). For example, amongst ToxA-like effector templates, the model with the lowest REU value of −230.0 had a TM-score of 0.45, whilst the highest TM-score was 0.52 (Figure 3A), and the lowest REU of −190 had a TM-score of 0.45, with the highest TM-score of 0.65 (Figure 3C). The lowest REU of −209 had a TM-score of 0.36, whilst the highest TM-score was 0.42 (Figure 3B). This trend is also observed in the MAX effector templates (Figure 4). The lowest REU values were −138 (Figure 4C), −134 (Figure 4F), and −110 (Figure 4E), and had corresponding TM-scores of 0.28 (the highest was 0.4), 0.37 (the highest was 0.49) and 0.33 (the highest was 0.57), respectively. REU values of −97 (Figure 4B), −89 (Figure 4D), and −87 (Figure 4A) corresponded to TM-scores of 0.55 (the highest was 0.9), 0.35 (the highest was 0.48), and 0.36 (the highest was 0.49), respectively. These observations suggest that REU is not a reliable indicator of the ab initio model closest to the native structure.

We next determined whether a low RMSD value corresponds to the model closest to the native structure. Ab initio models with the lowest RMSD value for each template were superimposed with their respective native structures to evaluate their structural similarity (Appendix A in the Appendix A). The structures with the lowest RMSD to the template did not show any similarity to the native structure, with the exception of AvrPib. This is due to the fact that AvrPib has fewer loop regions in its structure compared to other effector templates (this is further validated below). Most of the ab initio models with the lowest RMSD exhibited incomplete secondary structure folds, and this was reflected in their average REU values. RMSD does not appear to be the best criterion for identifying the best ab initio model, even though RMSD calculates the average distance of Cα atoms between two protein structures and is widely used as a reliable indicator of their mutual structural deviation.

The final assessment determined whether high TM-score values correspond to the ab initio models closest to the native structure. Ab initio models with the highest TM-score values for each template were superimposed onto their respective native structures to evaluate the structural similarity (Figure 3 and Figure 4). The models with the highest TM-scores were found to have the closest structural similarity to their native structures. This indicates that, unlike RMSD, TM-scores of above 0.5 (which indicates structural similarity) can be used reliably to discriminate a good model from the pool of models generated by Rosetta.

Since the use of TM-score can better identify global fold similarity, the term ‘best model’ will be used henceforth to describe predicted models with the best TM-score (the highest value) for each effector template. The characteristics of the best structural model for each effector template are summarised in Table 1. It is important to point out that the best model identified will not necessarily be correct or similar to the template structure, especially if the TM-score value is below 0.5. This can indeed be observed for some of the effectors with a TM-score below 0.5 (such as in AvrPik, Avr2 (*Fol*) and Avr1CO39) in Figure 3 and Figure 4, but which were still deemed to be the best models since they were the best-predicted structures for those specific effector templates. Table 1 also reveals discrimination of the best model on the basis of TM-score, except for the best overall model, AvrPib (discussed further below).

### 2.2. Selection of the “Best” Predicted Models of ToxA-like and MAX Effector Proteins

Within the ToxA-like family, the AvrL567 effector was predicted to have the best model as it had the best TM-score amongst other effectors, exhibiting a similar β-sandwich fold with a complete set of seven β-strands when compared to the template structure (PDB ID: 2QVT). The second-best-predicted model was obtained for the ToxA effector with a slightly lower TM-score. Compared to the template structure (PDB ID: 1ZLD), the predicted ToxA model contained all seven β-strands, but these were oriented differently, such that a β-sandwich fold is formed by three- and four-strand β-sheets. The predicted model consists of the four-strand β-sheet as in the template, whilst the remaining three β-strands also form a β-sheet that is oriented in a perpendicular manner (Figure 5A). The 3_10_-helix at the N-terminus is also missing in the predicted model, and two new short α-helices are predicted in the C-terminal region. Amongst the ToxA-like effectors, the best model for Avr2 (*Fol*) was, in fact, the worst prediction since it does not resemble the template structure (PDB ID: 5OD4), with a TM-score below 0.5, indicating that it has no significant similarity to its template, unlike the best models for AvrL567 and ToxA (Figure 5B). This model lacks the β-sandwich fold and consists of only three β-strands (instead of seven) close to the core region of the template structure. A long α-helix was also wrongly predicted in the N-terminal region compared to the template structure, in which it is a disordered region anchored to the core of the β-sandwich by a disulfide bond (Cys40-Cys130). The predicted models of ToxA-like effector, Avr2 (*Fol*) and other MAX effector templates (ToxB, AvrPik, Avr1CO39, AvrPia and AvrPiz-t) that contain disulfide bridges were, as expected, not fully consistent in these regions with the crystal structures. This is mainly due to the models lacking secondary structure predicted in the regions involved in disulfide bridge formation, which affects the overall conformation in that region. Most of the disulfide bridges in the crystal structures of the effectors are formed between a cysteine located at the start of a β-strand and a cysteine located in a loop region, resulting in deviations between the pairing cysteines. The distances between the paired cysteines in the predicted models were predominantly 5–10 Å, potentially allowing additional modelling to be undertaken to re-introduce the disulfide bridges.

In the case of MAX effector templates, the best-predicted model for AvrPib was of very high quality and had the best overall TM-score (0.923) amongst all other predicted MAX effector protein models (Figure 6B), closely resembling the structural template (PDB ID: 5Z1V). This is also reflected by a low RMSD value of 0.7 Å when superimposed to the template structure, which in this particular case shows that RMSD can be a useful comparative metric to indicate structural similarity for effector proteins with fewer loop regions within their global structural fold, such as AvrPib. The best model predicted has five β-strands and is missing one short β-strand (of only two residues) in the N-terminal region. The next best-predicted models are those for AvrPia and AvrPiz-t, with average quality with TM-scores of 0.5639 and 0.51348, respectively. The best models with TM-score values between 0.5 and 0.6 exhibit structural defects, such as missing secondary structures, despite still maintaining the correct overall fold.

In the case of AvrPia, three β-strands are predicted instead of five, but only two of the β-strands match the template structure (PDB ID: 2N37) (Figure 6E). The best model for AvrPiz-t was predicted to have four β-strands, similar to the template structure (PDB ID: 2LW6), but is missing two short β-strands (two residues long) at the N-terminal region (Figure 6F). The best model for ToxB was of relatively lower quality, with a TM-score of 0.5022. This model has three of the total six β-strands missing compared to the template structure (PDB ID: 2MM0) (Figure 6A). In the case of AvrPik and Avr1CO39, their best models have low TM-scores (both below 0.5). The best model for AvrPik lacks the β-sandwich structure, with only two β-strands predicted, and the rest of the structure is disordered, with three α-helices (Figure 6C) compared to the template structure (PDB ID: 6FUB). The best model for Avr1CO39 is missing two of the total six β-strands, which are instead predicted as disordered loops, and contains an α-helix at the C-terminal (Figure 6D) compared to the template structure (PDB ID: 5ZNG).

### 2.3. Clustering of Structural Models of ToxA-like and MAX Effectors

The energy landscape of protein folding is extremely rugged, and in the absence of adequate conformational sampling, there is no guarantee that the global minimum will be found. In this study, Rosetta ab initio modelling was used to generate up to 50,000 independent models, and TM-score values were determined to be the most reliable metric to discriminate between good (near-native) and bad (incorrect) models, given that structural templates are available for each effector. Since good structures account for less than 10% of the models predicted by Rosetta, clustering was used to group together similar structural conformations within the pool of predicted models, followed by an assessment of the largest conformational clusters of all predicted ab initio models, based on the assumption that the lower energy minima are often the most sampled ones. Consequently, a combination of conformational clustering and energy ranking was used to identify the most native-like structure (or the best model, as described above) of ToxA-like and MAX effectors amongst the largest clusters. Clustering was performed using TM-score with the MaxCluster method, which resulted in a more efficient way of clustering predicted models compared to the commonly used RMSD clustering method since the best model for each effector could be found amongst the top largest clusters.

Only 10% of the models with the best REU were retained for clustering because most predicted models with the best TM-scores were found within the top 10% best REU. Additionally, reducing the clustering sample substantially reduced the computational cost and increased the clustering efficiency in terms of the probability of finding the native-like model among the top largest clusters. The main reason for finding or retaining the best model within the largest cluster is to later apply this approach to identify the ‘best model’ for structurally unconfirmed effectors, which obviously do not have any template structure available for comparison (i.e., to identify a good or bad structure based on TM-score or RMSD values). If clustering is unsuccessful, the best model will not be found in the largest cluster, which will contain incorrect models, making this approach unviable.

The best models for seven out of the nine effector proteins of both ToxA-like and MAX families were found within the top 10 largest clusters (Table 2). The exceptions are AvrPia and AvrPiz-t, whose best models were not found within the 10% best REU, making them unavailable during clustering. The best models for one of the ToxA-like effectors (Avr2 (*Fol*)) and three of the MAX effectors (ToxB, AvrPik and Avr1CO39) were found in the corresponding largest clusters, indicating that the MaxCluster method succeeded in clustering the effector models effectively.

We also tested the efficiency of the clustering method with the predicted models obtained from different *nstruct* values for each of the effector proteins. This was undertaken to determine if the results were consistent, such that the best model for each effector for each *nstruct* value was retained in the largest cluster. The identification of the best model for each effector upon clustering with different values of *nstruct* is summarised in Table 3. The best models predicted for all effector templates were found within the top 10 largest clusters, except for AvrPia, whose best model was never found within the top 10 largest clusters across all *nstruct* values. Some of the cluster ranks (indicated as NA in Table 3) did not have the best model within the 10% best REU, preventing it from being included in the clustering step. It can be concluded that MaxCluster clustering was successful in clustering the best models within the top 10 largest clusters for most of the effector templates.

Since most of the best structural models for all effector templates were found among the top 10 largest clusters, this approach for identifying the best model was applied to the selection of the structural models of the structurally unconfirmed effectors. For this, the top 10 final models were taken from the centroids of the top 10 largest clusters, which assumed that one of the clusters contained the best model. This approach is required since the selection of the best structural model for the structurally unconfirmed effectors will solely rely on the clustering step in order to select the correct model because no template structures are available for comparison (and, hence, TM-scores cannot be computed). In order to minimise the number of final models that need to be considered, one approach would be to select the largest cluster but is not warranted by our own findings. The clustering statistics for ToxA-like and MAX effectors reported in Table 3 show the rank of the best model using different *nstruct* values, revealing that only 12 out of a total of 45 best models (nine effectors × five different *nstruct* values = 45) were found to belong to the largest cluster. In the case of ToxA and AvrPib, none of the best models were found in the largest cluster, regardless of the value of *nstruct*; nevertheless, the best models were still within the top 10 largest clusters. The best model for AvrPia was ranked within the top 20 largest clusters (for an *nstruct* value of 10,000), whilst the best model for Avr1CO39 was ranked within the top 13 largest clusters (for an *nstruct* value of 40,000). The majority of the best models for AvrPia with *nstruct* values other than 10,000 could not be selected since they were not within the 10% best REU. It was also observed that for effectors where the best model was not found in the largest cluster, the centroid models of the largest clusters were incorrect (with TM-scores below 0.5), such that they could not be considered correct models.

We then proceeded to try to identify the best (correct) model for the structurally unconfirmed effectors by selecting the top 10 largest clusters, followed by the further selection of the best model on the basis of identifying structural homologues using DALI as well as matches to known superfamilies or folds in the SCOP and CATH databases. With the use of the MaxCluster nearest neighbour clustering method, the data in Table 3 suggest that the use of *nstruct* values of 30,000 and 50,000 provides the most extensive coverage of clusters (i.e., where the best structural models belong to the top 10 largest clusters) within the 10% best REU. By contrast, an *nstruct* value of 20,000 results in the lowest coverage, i.e., the smallest number of effector templates (ToxA, Avr2 (*Fol*), ToxB and AvrPib). This is followed by an *nstruct* value of 40,000 (with five effector templates), 10,000 (with six effector templates), and then both 30,000/50,000 with seven effector templates. This ranking of *nstruct* values is also consistent with the best model belonging to the top cluster (the largest cluster): 20,000 with only Avr2 (*Fol*) having the best model in the first cluster, followed by 40,000 and 10,000 with two effectors each, 30,000 with three effectors (Avr2 (*Fol*), AvrPik and Avr1CO39), and 50,000 with four effectors (ToxB, AvrPik, Avr1CO39 and AvrPiz-t). Theoretically, 50,000 would have been the preferred *nstruct* value due to more extensive sampling of conformations in Rosetta and resulting in the largest number of best models (four) ranked within the top cluster compared to other *nstruct* values, i.e., two for 40,000, three for 30,000, one for 20,000 and two for 10,000 (Table 3). However, given the computational cost of running Rosetta ab initio modelling, an *nstruct* value of 30,000 was selected as the optimum *nstruct* in this method and subsequent clustering of the predicted models.

### 2.4. Cross-Validation of Ab Initio Rosetta Models with QUARK and Robetta

Comparisons of the best structural models predicted from different tools were made on the basis of the superimposition of the models with their respective template structures and evaluation of the Cα RMSD and TM-score values. For the ToxA-like effector family, the best models generated with QUARK were found to be the closest to the native structure with the highest TM-score value, followed by those generated by Rosetta and Robetta (Figure 7).

QUARK was found to perform better with ToxA and AvrL567 effector proteins compared to Rosetta, with TM-scores of 0.8736 and 0.92977, respectively (Figure 7A,C). The best models predicted with Robetta had the lowest overall TM-scores for all ToxA-like effectors, ranging from 0.36213 to 0.5, and were clearly not reasonable.

In the case of the MAX effector family, the best models generated with Rosetta were found to be the closest to the native structure (with the highest TM-score values), followed by those generated by QUARK and Robetta (Figure 8). Rosetta ab initio modelling generated models with the best overall TM-scores, with the highest being AvrPib with a TM-score of 0.9230. The exception was AvrPik, where QUARK produced the best model with a TM-score of 0.8941 compared to 0.51348 in Rosetta (Figure 8F).

Overall, QUARK ranked the best for predicting ToxA-like structures, whilst Rosetta ab initio modelling ranked the best for predicting MAX effector structures. Robetta was found to perform below average in the modelling of ToxA-like and MAX effector structures in spite of generating the best models with the lowest TM-scores compared with the other two methods.

### 2.5. Modelling of Fungal Effectors Lacking Known Structures

The Rosetta ab initio modelling approach described above was used to predict the structures of structurally unconfirmed effector proteins as listed in Appendix A, and here the 10 best-predicted effector models: Avr2 (*F. fulva*), AvrStb6, BAS2, MiSSP7, BAS4, PstSCR1, ECP1, AVRA1, Iug6 and PstPEC6 were reported. The final model for each one of these structurally unconfirmed effectors was identified from the centroids of the top 10 largest clusters obtained using MaxCluster analysis (see Appendix A in the Appendix A), which were scanned against all structures in the Protein Data Bank for any matching structural homologues using the DALI server. The predicted structures of the structurally unconfirmed effectors are shown in Figure 9. The final models were selected using the representatives of the top 10 largest clusters, which were submitted to the DALI server and selected based on having the largest DALI Z-score and the lowest DALI RMSD. Most of these structures consist of a mix of at least one α-helix at the N- or C-terminal regions and anti-parallel β-strands, except for Avr2 (*F. fulva*), which is predicted to have five β-strands (Figure 9A), and MiSSP7, which is predicted to have three α-helices (Figure 9D).

Interestingly, all 10 effector models (Figure 9) generated by Rosetta ab initio have a structural match with proteins in the PDB, validating the existence of the structural folds predicted by this ab initio approach. Based on the analysis of the structural homologue matches identified by the DALI server, all ab initio models had a DALI Z-score above 2 and RMSD below 4.0 Å (Table 4). The structural homologues of the effector proteins identified involved various proteins of bacterial, mammalian and plant origin, which can be pathogenicity- and non-pathogenicity related.

The structural homologues of the predicted models for AvrStb6 of *Zymoseptoria tritici* and PstSCR1 of *Puccinia striiformis* were both pathogenicity-related proteins in bacterial pathogens: type II secretion system protein J (T2SS) and type VI secretion system protein (T6SS), respectively. The predicted model for AvrStb6 consists of an α-helix at the N-terminus followed by three anti-parallel β-strands. The specific structural homologue refers to XcpV of T2SS, a minor pseudopilin and part of the core of the pseudopilus tip complex in *Pseudomonas aeruginosa*, which transports exoproteins out of cells and is involved in the recognition of secretion substrates. The predicted fold of AvrStb6 overlaps with the full domain of T2SS protein J (T2SSJ) in the 38–125 amino acid range, but the predicted short α-helix at the N-terminus does not overlap with the T2SS domain (Figure 10A). The full domain of T2SSJ consists of four anti-parallel β-sheets and one α-helix in the C-terminal region. In the case of PstSCR1, the predicted model consists of two α-helices and four β-strands. Its structural homologue is the immunity protein TplEi of T6SS protein from *Pseudomonas aeruginosa*. PstSCR1 overlaps with the N-terminal domain of Tle cognate immunity protein 4 at amino acid positions 32–167 but not with the C-terminal domain (Figure 10B). The N-terminal domain consists of an anti-parallel β-sheet sandwiched by two α-helices and a short anti-parallel β-sheet.

Three structurally unconfirmed effector proteins, Avr2 of *Fulvia fulva*, AVRA1 of *Blumeria graminis f. sp. hordie* and PstPEC6 of *Puccinia striiformis*, were found to have structural homologues with bacterial proteins that may be indirectly related to pathogenicity. The predicted model for Avr2 (*F. fulva*) has five anti-parallel β-strands and has a structural homologue with the outer surface protein A/B, a lipoprotein in the bacterial pathogen of Lyme’s disease, *Borrelia burgdorferi sensu lato*. The predicted model for AVRA1 has five anti-parallel β-strands and three α-helices and has a structural homologue with a lipoprotein of *Salmonella*, a DUF1795 domain-containing protein. The predicted model for PstPEC6 has three anti-parallel β-strands, two π-helices and α-helix, and has structural homologues with a bacterial intracellular signalling protein consisting of DUF1696 with a pleckstrin–homology domain, UDP-N-acetylmuramoylalanine-D-glutamyl-lysine-D-A, an enzyme involved in the formation of the bacterial cell wall that plays a role in bacterial resistance.

The remaining five effectors (BAS2, MiSSP7, BAS4, ECP1 and Iug6) were found to have structural homologues with proteins that are not identified as being involved in plant pathogenicity. BAS2 has a structural homologue with the serine–threonine protein kinase involved in centriole assembly. MiSSP7 has a structural homologue with deneddylase, an enzyme found in the tegument of the herpes virus. BAS4 has a structural homologue with the rod-shaped determining protein MREB, which is an actin involved in bacterial shape maintenance. ECP1 has a structural homologue with the mediator of RNA polymerase II transcription subunit, which plays a role as mediator and kinetochore. Iug6 has structural homologues with the fermitin family homologue 1, which is a protein involved in cell adhesion assembly, as well as with neurobeachin, a human adaptor protein.

The ab initio models were also assessed for similarity with known protein folds and superfamilies in the CATH database to obtain more information about associated domains. The top CATH hits for each ab initio model are listed in Table 5. The CATH assessment (based on CATH classification ID) shows variations in the domain superfamilies of each structurally unconfirmed effector, which correlates with the fact that they are all evolutionarily unrelated. The majority of the ab initio models belong to the alpha–beta class: AvrStb6, BAS2, BAS4, PstSCR1, AVRA1 and Iug6. A mainly beta class were observed for Avr2 (*F. fulva*) and PstPEC6, whilst a mainly alpha class was observed for MiSSP7.

### 2.6. Structural Quality Assessment of Ab Initio Models

The best models for ToxA-like, MAX and the structurally unconfirmed effector proteins generated using Rosetta ab initio modelling were evaluated for their structural quality using PROCHECK, Verify3D and ProSA (Appendix A). PROCHECK performs several evaluations, including psi/phi angles in Ramachandran plots, where a good quality model is expected to possess over 90% of its amino acid residues in the most favoured regions. The Ramachandran plot analysis for the final models of four effectors (AvrPib, MiSSP7, BAS4 and Iug6) had more than 90% of their residues in the most favoured region, whilst two effectors (ToxB and Avr1CO39) had less than 80% with 79% and 77%, respectively. Apart from AvrL567 (with 0.9%), none of the effectors were found to possess residues in disallowed regions.

Based on Verify3D analysis, a reliable ab initio model should have more than 80% of residues with scores above 0.2. Eight ab initio models (AvrPib, AvrPik, AvrStb6, BAS2, ECP1, AVRA1, Iug6 and PstPEC6) had more than 80% of their residues with an overall average score of more than 0.2, which indicates that they are of acceptable quality. Based on ProSA analysis, a Z-score of −5.0 indicates that the structure is close to the NMR or XRD quality. The ab initio models were found to have Z-scores in the range of −1.13 to −5.82.

On the basis of the quality assessment analysis, we can conclude in general that the majority of the ab initio models are of acceptable quality. The quality assessments constitute general indicators, such that if a model does not meet the cut-off in one tool, it does not necessarily mean that it is of poor quality unless all three tools agree.

## 3. Discussion

We have described in detail our approach to predicting the structures of ToxA-like, MAX and structurally unconfirmed effector proteins using Rosetta ab initio modelling. The benchmarking studies showed that ab initio modelling could predict, with reasonable similarity, the overall structure and fold of some of the effectors from both the ToxA-like and MAX families. We also show that modelling structurally unconfirmed effectors can provide useful information in the form of structural matches to known protein folds with pathogenically relevant functions. This structure-guided approach may accelerate the identification of conserved effector folds and their interacting host resistance proteins [38], of which the latter can be engineered in the design of novel resistance plant proteins [50,51,52] for crop protection. We now centre the discussion of our findings on two themes. The first is our approach for the prediction of the structures of ToxA-like and MAX effector families, which includes the use of the Rosetta energy function, the impact of long loop regions, the presence of specific secondary structure folds, and the use of short fragment structures. The second is the application of our approach to predict the structure of structurally unconfirmed effector proteins and the possibility of identifying structural folds not previously observed in effector structural families.

We have observed, in general, that Rosetta ab initio models of effector templates and structurally unconfirmed effectors are robust and reliable and which could be validated structurally and, potentially, functionally. A protein model with the best (i.e., lowest) REU is usually the one with the largest amount of secondary structure content, although this does not necessarily mean that the correct fold has been predicted successfully. A metric such as RMSD can be used to compare the ab initio model with a template structure, although the model with the lowest RMSD is not always the best model because most effector protein structures contain long disordered loops, which are unique structural features of fungal effector proteins, including ToxA-like and MAX effectors. Since TM-score compares structures based on global topology and is less sensitive to local structure variations (loop regions), TM-score values of the target with respect to the template structure were thus used to determine the similarity of the predicted ab initio models with available templates and to develop an approach to select the best-predicted structural models for structurally unconfirmed effectors where no RMSD or TM-score is available. Recently published work on the prediction of effector protein structures successfully applied TM-score (pTM) to measure structural similarity [53]. In general, TM-score has been proven to show a strong correlation to the quality of the predicted model when compared to the template structure [54] (in this case, the effector protein templates) and is used as a reliable assessment of protein structure prediction in CASP [55] to differentiate a good model from a bad one.

### 3.1. Secondary Structure Characteristics of Effector Proteins and Accuracy of the Predicted Models

Several factors were identified that affect the quality of the predicted structures of the fungal effector proteins considered in this study. The first one is the length and type of the secondary structures in ToxA-like and MAX effector proteins. Although both exhibit a similar β-sandwich fold, different levels of accuracy in the predicted models were found to be attributed to these structural families having differences in average protein length and the type of secondary structures (α-helices and β-strands), which Rosetta ab initio modelling is sensitive to. Previous Rosetta simulation studies have revealed that the ability to accurately model the structure of a protein is highly dependent on the length of helices, strands and loops [56]. In this study, a strong correlation was observed between the accuracy of the predicted models and the arrangement and number of α-helices and β-strands, as well as the length of loops. This could be related to the percentage content of long loop regions and the accuracy of the predicted type and alignment of these loops, which could favour specific tertiary structure motifs. This was the case with the best models for Avr2 (*Fol*) and AvrPik, which had poor TM-scores of 0.42148 and 0.39585, respectively. Since Rosetta has been parameterised with experimental structures, we speculate that most of the above inaccuracies in the predicted models arise from flaws in the prediction of such long loops. The errors that arise from the alignment of loops are indeed a well-known limitation of structural predictions. Previous studies about the diversity of these loops have found that the length, anchor region and even local and non-local interactions within these regions can favour specific secondary structural patterns, which has an impact on the predicted final model [56,57,58].

Rosetta ab initio modelling is known to exhibit a preference for predicting α-helices in regions that are known to have no secondary structure. For example, the model of AvrPik was predicted to have α-helices in regions that are composed of long disordered loops in the template structure (Figure 6C). Another α-helix in the N-terminal region was also predicted to be longer than is found in the template structure. The model of Avr2 (*Fol*) was predicted to have an α-helix instead of the first β-strand in the template structure (Figure 5B). In addition, there are also several predicted short 3_10_-helices that occupy regions that are long disordered loops in the template structure. This might also be due to the preference of Rosetta ab initio energy functions for α-helices, as well as the length of the fragment structures used in this study, since the use of 9-mers only biases the generation of α-helical structures, whereas the use of 3-mers biases for both α-helices and β-strands [57]. In addition, modelling helical regions tends to be easier in protein folds with high helical content because of their lower structural diversity compared to folds with high content of β-strands and loops. This is also known to be related to the lack of sufficient accuracy in the Rosetta energy score function for assessing β-strands [59]. This is probably due to the use of a flexible backbone in the Rosetta relax protocol since the use of fixed-backbone calculations seems to be better for the evaluation of β-sheets compared to α-helices [60]. For example, this is the case with AvrPiz-t, which consists of an N-terminal tail composed of two short β-strands, two residues long each (Figure 6F). The short β-strands (approximately two residues long) located within a long-disordered loop in AvrPiz-t were predicted to have α-helical conformation; however, it is not clear if there is a minimum number of residues required for the prediction of β-strand formation. Furthermore, a high ratio of disordered loops to short β-strands may lead to a bias towards the prediction of α-helices over β-strands.

The second factor limiting the quality of the predicted structures relates to the content of disordered regions or long loops present in the protein structure, which are unique structural features found in most effector proteins within the same structural family. Rosetta ab initio appeared to be unable to predict a reliable model if the protein has a relatively high content of disordered regions. This is the case, for example, with Avr2 (*Fol*), AvrPik and Avr1CO39 (Figure 5B, Figure 6C,D, respectively), all of which have TM-scores below 0.5. Based on the DSSP analysis of the experimental 3D structures of ToxA-like and MAX effectors (Appendix A), generally, more than half of all amino acid residues were found to be disordered at the N- and C-termini, as well as loops and turns of different length. Disordered regions in proteins are known to be involved in regulation and signalling [61] and can play a functional role in protein–protein interactions, adopting a well-defined conformation upon interaction with a specific binding partner [62]. The prediction of structures with high content of disordered regions is challenging for Rosetta because ab initio modelling searches for the lowest energy conformation, and a proper estimate of the contribution of disordered regions to the free energy requires an accurate computation of entropy and energy [63]. In Rosetta, disordered regions only make repulsive, unfavourable contributions to the total energy. A correlation between the number of disordered regions and the model quality or energy score has been reported, with scores being less favourable if the protein is largely disordered [64]. Based on this study, we can conclude that effector proteins with few or no disordered regions can be modelled accurately, and this is crucial in determining beforehand the likely success of predicting reliable models of structurally unconfirmed effector protein structures. The presence of disordered regions or flexible surface residues can indeed be predicted using tools such as DISOPRED, DISEMBL and PROFbval [65,66].

### 3.2. Comparison with Other Ab Initio Methods

Since ab initio modelling is non-template-based, comparison with models generated using other non-template-based modelling methods can increase confidence in the accuracy of the predicted models. In this study, the structural models generated by Rosetta for the set of ToxA-like and MAX effectors were compared with those generated by QUARK and Robetta. Overall, QUARK was found to be the best method for modelling ToxA-like structures, whilst Rosetta ab initio modelling was found to be the best method for modelling MAX effector structures. Robetta was found to perform the worst in the modelling of both ToxA-like and MAX effector structural families generating final models with the lowest TM-scores among all methods.

The success of QUARK in modelling ToxA-like effectors may be due to the utilisation of structural fragments with varying lengths between 1 and 20 residues during modelling [67]. The use of multiple fragment lengths has been shown to generate better ab initio models, with longer fragment lengths (20 residues) being more useful during the early stages of modelling [68]. Such optimisation was not conducted in our study with Rosetta ab initio modelling since only default 3- and 9-mer structural fragments were used. In addition, the content of α-helices, β-strands and loop regions in effector proteins should also be taken into account when deciding what the appropriate length of structural fragments should be. Shorter fragments have been shown to be more efficient for modelling structures with higher β-strand content, whilst longer fragments are better for modelling structures with higher α-helical content [68].

In this study, up to 50,000 models per run were generated with Rosetta, but this high number of models is not feasible with the Robetta webserver due to the high computational cost involved. Consequently, Robetta uses an *nstruct* value of 1000; however, the low number of models produced may limit the sampling of potential folds.

Robetta applies Rosetta’s legacy clustering to identify similar structural models based on RMSD cut-off during conformational clustering. Whilst this approach is commonly used, the unique structural features of ToxA-like and MAX effector proteins discussed earlier may result in models with similar folds being incorrectly grouped in separate conformational clusters because of the presence of long, disordered loops. This may lead to the grouping of good and bad models in the same cluster, and the best model may not be present in the largest cluster. This makes RMSD clustering an unviable approach for the structurally unconfirmed effectors and has been overcome in this study by implementing the MaxCluster nearest neighbouring method, which uses TM-scores instead of RMSD. This approach eliminated the problem of clustering models that had significant disordered regions. Consequently, structural models with similar folds, regardless of the presence of long disordered loops, could be clustered effectively, resulting in the best model being identified within the top 10 largest clusters. Since MaxCluster clustering could be used successfully to separate good models from bad ones, the same approach was used for clustering the structural models of structurally unconfirmed effectors. This customised protocol was incorporated in the ab initio modelling of structurally unconfirmed effector proteins.

A comparison of the application of Rosetta with the recently released deep learning ab initio modelling program AlphaFold (collaborative version) was also included in this study for structurally unconfirmed effectors (Appendix A). Overall, structures modelled using AlphaFold exhibited long disordered loops in regions that were modelled as α-helices in Rosetta. Examples of this are effectors AvrStb6, MiSSP7, BAS4, PstSCR1 and Iug6. A more compact, folded structure was obtained for effector AVRA1 using AlphaFold, but the remaining effectors were better modelled with Rosetta. The application of AlphaFold to the structural modelling of fungal effectors may still require optimisation even though this program performs well for the majority of proteins, but there may still be issues for proteins that contain low numbers of intra-chain contacts or homotypic contacts [69]. Nevertheless, improved structural prediction methods such as AlphaFold will advance the use of ab initio modelling for the prediction of fungal effector protein structures [39] and will continue to improve with growing numbers of experimentally determined effector structures [52].

### 3.3. Structural Homology Detected for Predicted Structures of Structurally Unconfirmed Fungal Effectors

We predicted structures of phenotypically validated effector proteins obtained from PHI-base [70] for Avr2 (*F. fulva*), AvrStb6, BAS2, MiSSP7, BAS4, PstSCR1, ECP1, AVRA1, Iug6 and PstPEC6. These effectors are not known to be structurally related and likely play different roles in fungal pathogenicity [41,42,43,44,45,46,47,48,49,71]. The selection of their final representative models was carried out on the basis of identifying matches with known PDB structures and known folds and superfamilies in the CATH and SCOP databases. Whilst structural homologues could be identified for these phenotypically validated effectors, as expected, no matching domains could be found using searches based on sequence homology. None of the ab initio models of these structurally unconfirmed effectors have matches with known fungal effector structural families. However, each effector was predicted to have different numbers of α-helices and β-strands within a reasonable and identifiable fold. Most of the β-strands were seen to form anti-parallel β-sheets, and variations in the length of loops were observed, which is a common unique feature of fungal effector proteins.

Two structurally unconfirmed effectors, AvrStb6 and PstSCR1, were predicted to be structurally homologous to bacterial type II and type VI secretion system (T2SS and T6SS) proteins. Both are cysteine-rich and apoplastic avirulence effectors of *Zymoseptoria tritici* [42,71] and *Puccinia striformis* f. sp. *tritici*, respectively [45]. Both are wheat pathogens but relatively distant taxonomically, belonging to the Ascomyceta and Basidiomycota phyla. Analysis of AvrStb6 alleles indicates diversifying selection at amino acid residues 34 and 60, which are its primary interaction residues. AvrStb6 also has 12 cysteine residues that likely form disulfide bonds; although these were not predicted as the ab initio modelling did not use any disulfide bond constraints; this did not prevent the prediction of useful structural homology information. T2SS is a transport protein and is required for the delivery of proteases or toxins into the extracellular environment [72], while T6SS is known to secrete numerous toxins involved in bacteria–bacteria competition [73]. The structural homology between XcpV (of the T2SS) and AvrStb6 may suggest a role in effector protein transport. XcpV is involved in the assembly of pseudopilus tips [74]. PstSCR1 has structural homology with TplEi (of the T6SS), and TplE1 neutralizes the toxic effect of the anti-bacterial lipolytic toxin TplE to protect bacteria from auto-intoxication [73].

The structurally unconfirmed effectors Avr2 (*F. fulva*), AVRA1 and PstPEC6 were predicted to have structural homologues with bacterial proteins with potential roles in pathogenicity. The remaining structurally unconfirmed effectors, BAS2, MiSSP7, BAS4, ECP1 and Iug6, were predicted to have structural homologues that are non-pathogenicity-related proteins. On the basis of the structural homologues, fold and superfamily matches identified using the DALI, CATH and SCOP databases, respectively, and it is likely that the predicted ab initio models are correct representations of the structures of all of these structurally unconfirmed effectors, which should be validated in future using XRD crystallography or NMR spectroscopy. Interestingly, these ab initio models increase the structural diversity of the structural families of fungal effector proteins.

## 4. Materials and Methods

### 4.1. Input Data and Pre-Modelling Steps

Two types of fungal effector protein datasets were used: (1) ToxA-like and MAX effector protein “templates” with known structures, and (2) “structurally unconfirmed” effectors which are experimentally confirmed effectors lacking known structures and not belonging to a known structural family. Effector templates consisted of ToxA-like and MAX effector protein sequences with available XRD and/or NMR structures in the Protein Data Bank (PDB). The effector template protein sequences were obtained from the Universal Protein Resource (UniProt) database in FASTA format (Appendix A in the Appendix A). Structurally unconfirmed effector sequences were obtained from the training datasets used in EffectorP v1 and v2 (Appendix A in the Appendix A) [11], which were obtained from the Pathogen–Host Interaction database (PHI-base) version 4.8 [70,75].

SignalP version 4.1 [76] was used to predict secretion signal peptides and cleavage sites to obtain mature protein sequences, which were used in all subsequent analyses [77]. The Robetta Fragment server (https://robetta.bakerlab.org/fragmentsubmit.jsp) (accessed on 11 February 2019) [78] was used, which applied the Ginzu protocol to generate 3- and 9-mer fragments (within recognised domains and a high probability to fold into a defined secondary structure conformation) for each protein sequence used during ab initio modelling [79]. Fragments generated for the modelling of effector templates were filtered out to omit fragments that were derived from existing PDB structures of any ToxA-like and MAX effector proteins used in this study. Information about the secondary structures and disordered regions of the effector templates was obtained from the full validation report for each PDB ID entry in the PDB webserver (Appendix A).

### 4.2. Structural Modelling

The structures of the effector templates and structurally unconfirmed effectors were modelled using Rosetta version 3.8 (local installation) in the ab initio modelling mode [32,80]. Rosetta operates on the basic principle that proteins fold to their lowest energy state, where the native structure conformation will possess the lowest Gibbs free energy. Rosetta utilises a knowledge-based and Monte Carlo annealing search strategy to guide the sampling of φ (phi), ψ (psi) and ω (omega) angles by randomly combining short fragments into the structural sequence to favour models with protein-like conformations and a global minimum on the energy landscape [32]. The resulting structural models were relaxed in torsional and cartesian space using the Rosetta Energy Function (REF15) [81], which was evaluated and parameterised based on protein structures in the PDB. Rosetta scores are provided in arbitrary Rosetta energy units (REU). The REU of each model is dependent on several energy terms, such as Lennard-Jones attractive and repulsive energies, a Gaussian implicit solvation energy, hydrogen bonds, disulfide bridges and torsion angles [81].

The modelling of the structure of effector templates was evaluated based on a range of numbers of model structures (*nstruct*) generated per run with various values between 1000 and 50,000. This was undertaken to determine the minimum *nstruct* that could generate the best ab initio model close to its native structure. Meanwhile, the modelling of the structure of structurally unconfirmed effectors indicated an *nstruct* value of 30,000 was optimal (see Results). The modelling of the structures of effector templates reported did not involve the use of experimentally confirmed disulfide bond constraints, ultimately because it would not be possible to apply such constraints to the modelling of effector candidates.

### 4.3. Analysis of Ab Initio Structural Models

Ab initio models of the structures of effector templates and structurally unconfirmed effectors were clustered using MaxCluster 3D-Jury version GCC 3.4.3 (http://www.sbg.bio.ic.ac.uk/~maxcluster/index.html) [82,83]. Only 10% of the models with the lowest REU were retained for clustering. MaxCluster 3D-Jury module uses the nearest neighbour method and enables the clustering of models using a so-called template modelling (TM) score. The TM-score compares structures based on global topology and is less sensitive to local structure variations compared to the use of root mean square deviation (RMSD). The TM-score values ranged between 0 and 1, where a score greater than 0.5 indicates that structures have the same fold, and a higher value indicates greater similarity [84]. In the clustering of the structurally unconfirmed effectors, centroids (i.e., the representative model of that cluster) of the top 10 largest clusters were reported as the top 10 final models. Each ab initio model was superimposed with its experimental structure to determine the Cα root mean square deviation (RMSD) using the MaxCluster pairwise alignment tool [85]. TM-score values between the model and the template were generated using TMalign [86].

### 4.4. Structural Quality Assessment and Validation

The geometrical and stereochemical properties of the top 10 ab initio models were assessed to evaluate their quality using several tools: PROCHECK [87], ProSA [88] and VERIFY-3D [89] webservers (all accessed on 16 January 2020). PROCHECK evaluates protein stereochemical quality based on ideal characteristics of high-resolution protein structures in the PDB [87]. ProSA estimates a quality index of a protein fold by calculating the statistical Z-score deviation of a protein structure with respect to high-resolution experimental structures in the PDB. It compares and analyses the overall quality of the structures using an energy function for each residue position to determine whether they are native-like [88]. Verify3D assesses the quality of a protein structure by analysing 3D structure-sequence compatibility and can be used to confirm the environment (secondary structures and polarity) of side chains in ab initio models, in which the packing quality of each residue is assessed by scoring the compatibility of the amino acid residues with their environment, with 3D-1D scores above 0.2 corresponding to acceptable side-chain environments [89]. The final models of the structurally unconfirmed effector were further analysed to identify structural homologues with currently available 3D structures in the PDB using DaliLite.v5 (DALI) [90,91]. DALI uses Z-score as the similarity matrix in which Z-scores are weighted by the lengths of protein chains, which is different from the matrix (TM-score) that was used above and is length-independent. A DALI Z-score above 2 corresponds to similar folds with a strong match if the sequence identity is more than 20% or depending on the sequence length for structurally unconfirmed effectors [92]. The final model from the top 10 was selected according to the highest DALI Z-score and the lowest DALI RMSD. All of the final models were assessed for their classifications of domain superfamilies based on the CATH-Plus (v4.2) [93,94] and SCOP2 databases [95,96,97]. All of the figures were analysed and generated using open-source PyMOL Open Source (Schrödinger, LLC, New York, NY, USA).

### 4.5. Comparison with QUARK and Robetta

For the purpose of cross-validating the methods described above, the ab initio models obtained with Rosetta for the ToxA-like and MAX effectors were compared with models separately generated using different ab initio tools: QUARK webserver https://zhanggroup.org/QUARK/ (accessed on 9 October 2019) [67,98] and Robetta webserver (https://robetta.bakerlab.org/) (accessed on 12 September 2019) [79,99]. QUARK uses a neural network approach to predict diverse structural features from fragments of continuous lengths of 1 to 20 residues based on replica exchange Monte Carlo (REMC) simulations and a knowledge-based force field [67,98]. A total of 5000 models were generated from low-temperature replicas, which were then clustered based on the Cα RMSD using SPICKER [100]. Meanwhile, the Robetta Structure Prediction webserver uses the modified Rosetta ab initio package [79], in which 1000 models were generated in each run followed by Cα RMSD clustering of the 10% lowest scoring models. Both QUARK and Robetta servers report centroids of the top five biggest clusters as their final models. The structural models generated with both servers were compared to their respective template structures based on the Cα RMSD and TM-score values. The model with the best TM-score was reported for each effector template.

## 5. Conclusions

Protein structure ab initio modelling is a useful approach for predicting fungal effector protein structures that do not have any structural templates for typical homology modelling and protein threading methods. This work provides a useful approach to the modelling of ToxA-like and MAX effectors, followed by the implementation of a customised approach to model the structure of structurally unconfirmed effector proteins. In particular, this study highlights the success of predicting the structure of ToxA-like and MAX effector proteins using Rosetta ab initio modelling and the use of TM-score to select reliable models that are close to the template structure. Ab initio modelling was shown to accurately predict the structures of AvrL567 from the ToxA-like family and ToxB and AvrPib from the MAX effector family. However, less accurate predictions for several members of these families were found to be caused by several factors, such as the presence of long disordered loops connecting β-strands, the length of the β-strands that are connected by such loops, and the presence of either 3_10_- or π-helices. Success in modelling effector structures was found to be affected by the overall protein fold, secondary structure content and loop regions, reflecting the unique common structural features found in ToxA-like and MAX effectors. Our findings thus emphasise the importance of identifying unique structural features for each effector protein, which might require the customisation of Rosetta ab initio modelling to obtain more accurate models.

Most of the parameters used in this study for Rosetta ab initio modelling were set to their default values, which suggests that the accuracy of models predicted could, in principle, be further improved by including evolutionary contact data from predicted sequence alignments and using residue–residue contact predictions (obtained from tools such as GREMLIN) as constraints [37]. However, accurate contact prediction would require a large number of aligned sequences, which is currently not possible for fungal effector proteins because there is a limited number of phenotypically validated effector sequences. Other improvements could include the use of multiple lengths of short structural fragments to guide the modelling, which has been shown to increase accuracy, as well as to customise the length of fragments based on the content of secondary structures (α-helices and β-strands) in the protein sequence of the structurally unconfirmed effectors. The use of tools, such as LRFragLib, have been shown to improve ab initio modelling by accurately identifying near-native fragments and creating high-quality fragment libraries [101].

The successful modelling of new fungal effectors may shed light on the existence of undiscovered structural families due to the presence of protein folds not observed in this type of protein. This may be achievable solely on the basis of their amino acid sequence as well as through the development of new ab initio protein modelling methods, such as the use of deep learning in the recently developed AlphaFold [69]. Since there is a limited number of experimentally determined fungal effector protein structures, ab initio modelling may expand the database of potential folds and structures in these proteins. This is particularly important given the vast amount of sequence data available for fungal genomes and very recent reports of numerous effector-like, sequence-dissimilar families with predicted structural similarities but lacking functional studies [30].

## Figures and Tables

**Figure 1 ijms-24-06262-f001:**
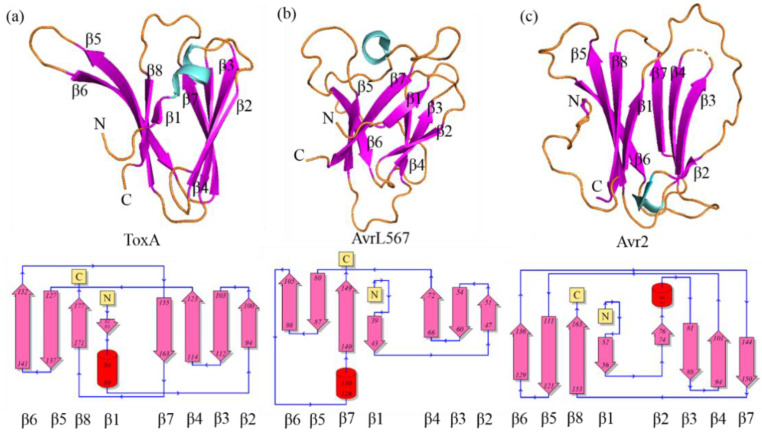
Structures of effector proteins from the ToxA-like family: (**a**) ToxA (PDB ID: 1ZLD), (**b**) AvrL567 (PDB ID: 2QVT) and (**c**) Avr2 (*Fol*) (PDB ID: 5OD4). The β-strands are shown in purple and the α-helix is shown in cyan. The structural topology of the secondary structures is shown below each corresponding structure. Figure was used under the terms of the Creative Commons Attribution License (CC-BY 4.0) [24].

**Figure 2 ijms-24-06262-f002:**
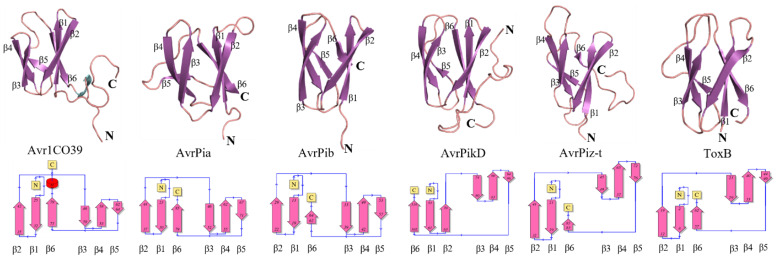
Structures of effector proteins from the MAX family: Avr1CO39 (PDB ID: 5ZNG), AvrPia (PDB ID: 2N37), AvrPib (PDB ID: 5Z1V), AvrPikD (PDB ID: 6FUB), AvrPiz-t (PDB ID: 2LW6) and ToxB (PDB ID: 2MM0). The β-strands are shown in purple, and the α-helix is shown in cyan. The structural topology of the secondary structures is shown below each protein.

**Figure 3 ijms-24-06262-f003:**
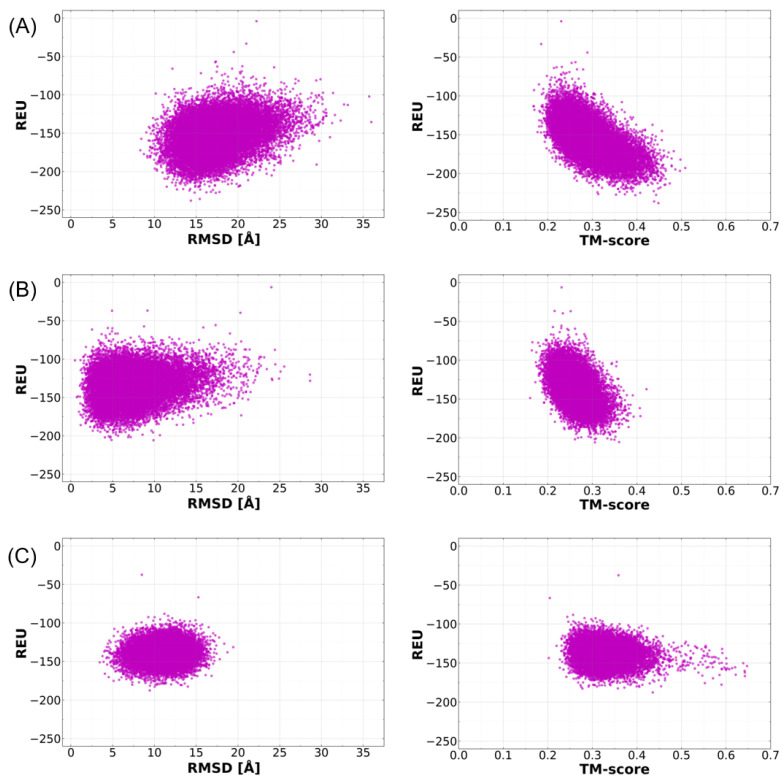
Scatter plots of the distribution of Rosetta energy scores (in Rosetta energy units (REU)) versus RMSD (**left panel**) and TM-score (**right panel**) of ToxA-like effector templates ToxA (**A**), Avr2 (*Fol*) (**B**), and AvrL567 (**C**). RMSD was calculated as the average deviation between the Cα of all residues in each model and the native structure (in Ångstrom). TM-score values above 0.5 indicate similarity in fold between the ab initio model and the native structure. Plots were derived based on structural models obtained with the highest *nstruct* of 50,000 models per run, and only models with negative REU values were included in the analysis.

**Figure 4 ijms-24-06262-f004:**
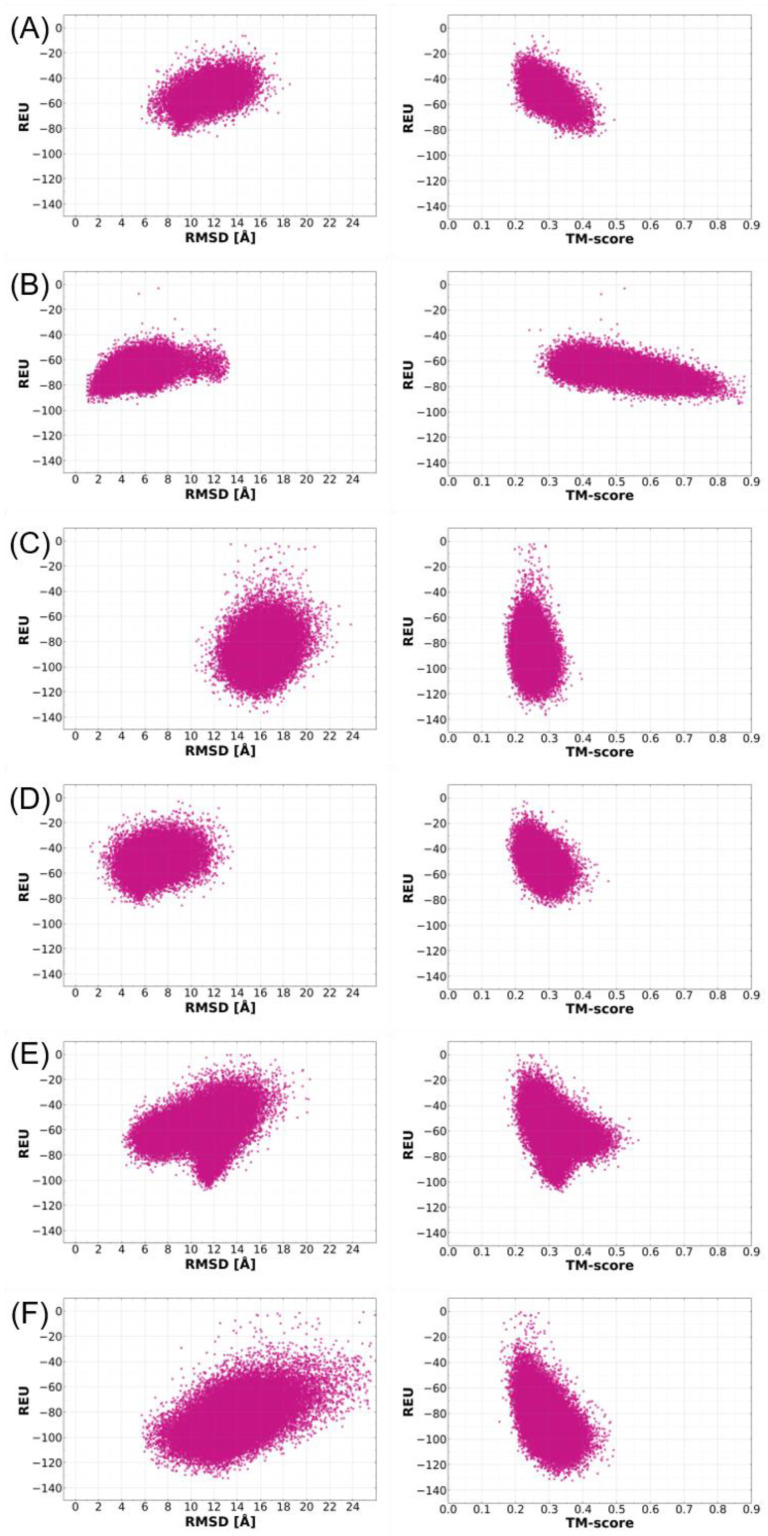
Scatter plots of the distribution of Rosetta energy scores (in Rosetta energy units (REU)) versus RMSD (**left panel**) and TM-score (**right panel**) of MAX effector templates ToxB (**A**), AvrPib (**B**), AvrPik (**C**), AvrCO39 (**D**), AvrPia (**E**), and AvrPiz-t (**F**). RMSD was calculated as the average deviation between the Cα of all residues in each model and the native structure (in Ångstrom). TM-score values above 0.5 indicate similarity in fold between the ab initio model and native structure. Plots were derived based on structural models obtained with the highest *nstruct* of 50,000 models per run, and only models with negative REU values were included in the analysis.

**Figure 5 ijms-24-06262-f005:**
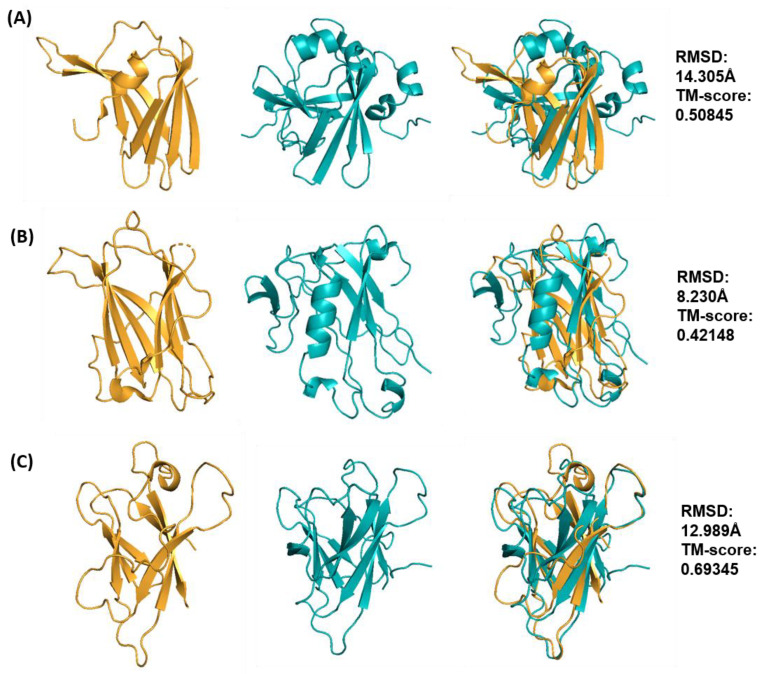
Best structural models for ToxA-like effector templates ToxA (**A**), Avr2 (*Fol*) (**B**), and AvrL567 (**C**) predicted with Rosetta ab initio modelling. All models are shown in cyan in ribbon representation and each model was superimposed onto their respective XRD or NMR template structure (shown in gold). RMSD and TM-score values are shown on the right-hand side.

**Figure 6 ijms-24-06262-f006:**
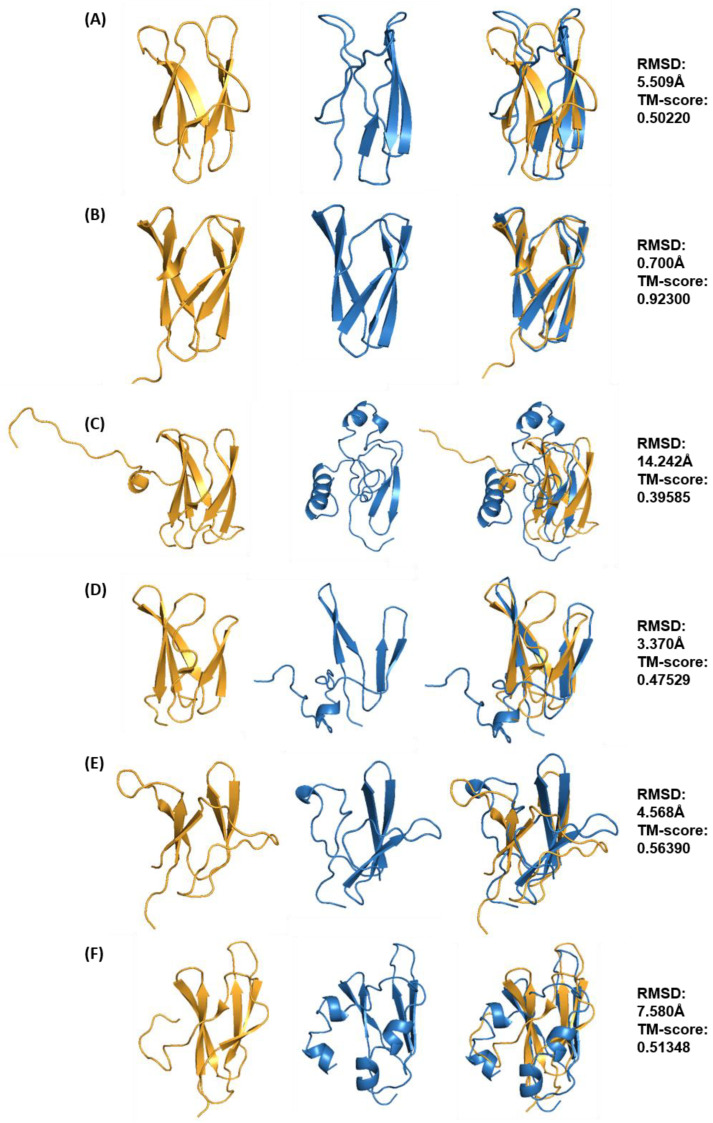
Best structural models for MAX effector templates ToxB (**A**), AvrPib (**B**), AvrPik (**C**), Avr1CO39 (**D**), AvrPia (**E**) and AvrPiz-t (**F**) predicted with Rosetta ab initio modelling. All models are shown in blue in ribbon representation, and each model was superimposed onto their respective XRD or NMR template structure (shown in gold). RMSD and TM-score values are shown on the right-hand side.

**Figure 7 ijms-24-06262-f007:**
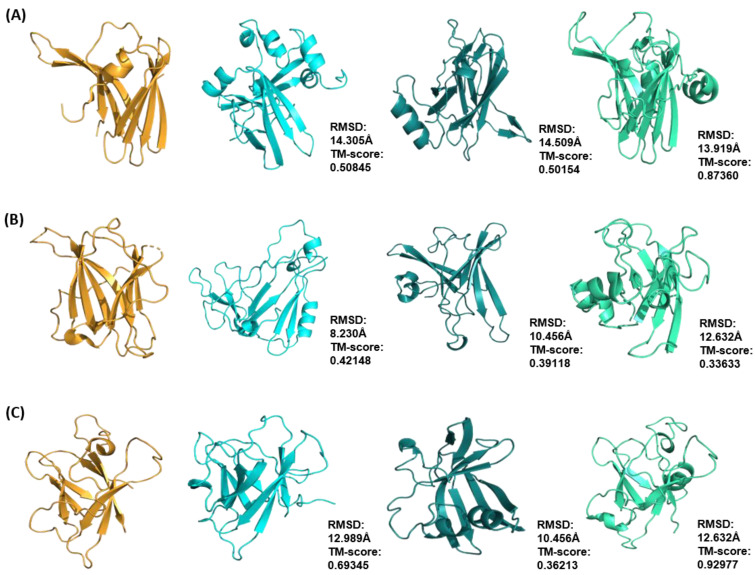
Comparison of Rosetta ab initio models of ToxA-like family (shown in cyan) ToxA (**A**), Avr2 (*Fol*) (**B**) and AvrL567 (**C**) with models generated with Robetta (shown in dark green) and QUARK (shown in light green). Each model was superimposed onto its respective XRD or NMR structure (shown in gold) and the corresponding RMSD and TM-score values are shown to the right-hand side of each model.

**Figure 8 ijms-24-06262-f008:**
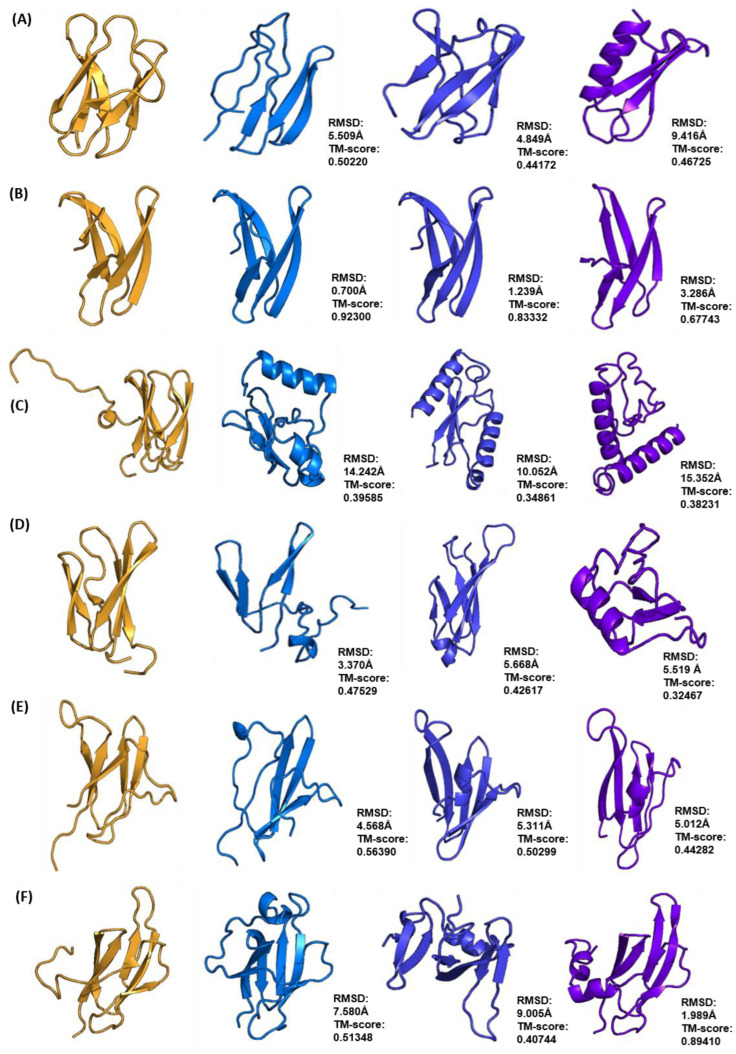
Comparison of Rosetta ab initio models of MAX family (shown in blue) ToxB (**A**), AvrPib (**B**), AvrPitz (**C**), Avr1CO39 (**D**), AvrPia (**E**) and AvrPik (**F**) with models generated with Robetta (shown in dark blue), and QUARK (shown in purple). Each model was superimposed onto its respective XRD or NMR structure (in gold) and the corresponding RMSD and TM-score values are shown to the right-hand side of each model.

**Figure 9 ijms-24-06262-f009:**
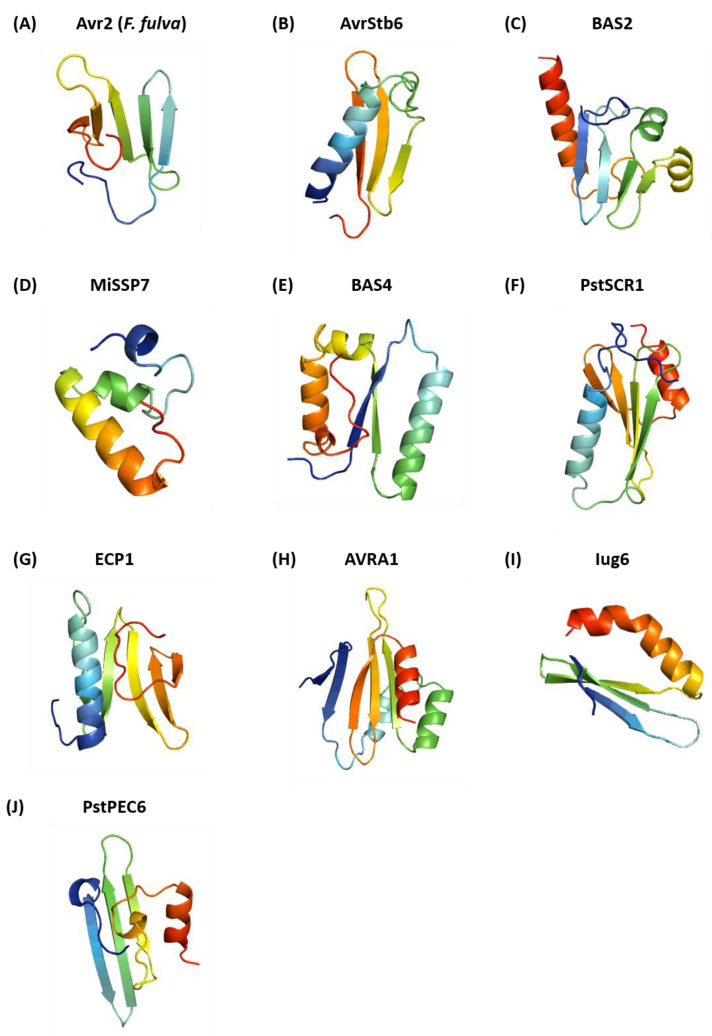
Rosetta ab initio models of structurally unconfirmed effector proteins obtained from PHI-base: Avr2 (*F. fulva*) (**A**), AvrStb6 (**B**), BAS2 (**C**), MiSSP7 (**D**), BAS4 (**E**), PstSCR1 (**F**), ECP1 (**G**), AVRA1 (**H**), Iug6 (**I**) and PstPEC6 (**J**). All models are represented with a rainbow colour scheme, from dark blue at the N-terminus to dark red at the C-terminus.

**Figure 10 ijms-24-06262-f010:**
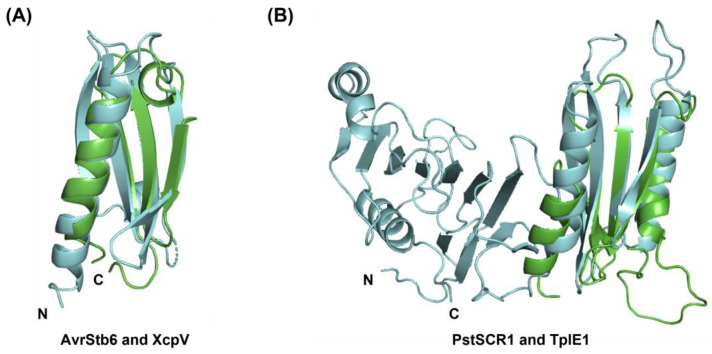
Superimposition of the final models of AvrStb6 (**A**) and PstSCR1 (**B**) with the structures of XcpV of T2SS and TplE1 of T6SS, respectively. Rosetta models are coloured in green and homologous structures in cyan.

**Table 1 ijms-24-06262-t001:** Characteristics of the best structural models obtained with Rosetta ab initio modelling for ToxA-like and MAX effector templates.

Effector Protein Template (PDB ID)	Species	TM-Score	RMSD (Å)	*nstruct*
ToxA (1ZLD)	*Pyrenophora tritici repentis*	0.50845	14.31	50,000
Avr2 (*Fol*) (5OD4)	*Fusarium oxysporum* f. sp. *lycopersici*	0.42148	8.23	40,000
AvrL567 (2QVT)	*Melampsora lini*	0.69345	12.99	30,000
ToxB (2MM0)	*Pyrenophora tritici repentis*	0.50220	5.51	40,000
AvrPib (5Z1V)	*Magnaporthe oryzae*	0.92300	0.70	10,000
AvrPik (6FUB)	*Magnaporthe oryzae*	0.39585	14.24	50,000
Avr1CO39 (5ZNG)	*Magnaporthe oryzae*	0.47529	3.37	50,000
AvrPia (2N37)	*Magnaporthe oryzae*	0.56390	4.57	50,000
AvrPiz-t (2LW6)	*Magnaporthe oryzae*	0.51348	7.58	7500

**Table 2 ijms-24-06262-t002:** Summary of clustering of the best structural models for ToxA-like and MAX effectors using the MaxCluster nearest neighbour method. The presence of the best structural model for each template was used as an indicator of successful clustering. The rank of the cluster that includes the best model and the total number of clusters for each effector are provided. Clustering was performed on the 10% best REU.

Effector Protein	Total Number of Clusters	Cluster Rank with Best Structural Model
ToxA	117	4
Avr2 (*Fol*)	59	1
AvrL567	57	7
ToxB	81	1
AvrPib	24	2
AvrPik	136	1
Avr1CO39	86	1
AvrPia *	137	NA
AvrPiz-t *	20	NA

* NA indicates that the best model is not included within 10% best REU.

**Table 3 ijms-24-06262-t003:** Summary of clustering of the best structural models for ToxA-like and MAX effectors using the MaxCluster nearest neighbour method for different *nstruct* values. The table provides the rank of the cluster that includes the best model and the total number of clusters. Effectors will have different best models for each *nstruct* value. Clustering was performed on the 10% best REU. Grey colour is used to highlight clusters that included the best model but were ranked outside of the top 10 largest clusters or where the best model was not within the 10% best REU and thus was not included in the clustering step (NA).

Effector Proteins	10,000	20,000	30,000	40,000	50,000
ToxA	2/33	2/41	4/57	2/90	4/117
Avr2 (*Fol*)	NA/23	1/25	1/46	1/59	NA/26
AvrL567	8/32	13/65	7/57	7/89	6/108
ToxB	1/21	8/39	3/62	1/81	1/96
AvrPib	2/24	3/47	8/71	3/91	2/110
AvrPik	1/40	NA/78	1/129	NA/105	1/136
Avr1CO39	NA/21	NA/40	1/84	13/56	1/86
AvrPia	20/28	NA/51	NA/28	NA/105	NA/137
AvrPiz-t	5/21	NA/47	NA/72	NA/95	1/128

NA indicates that the best model is not included within the 10% best REU.

**Table 4 ijms-24-06262-t004:** List of structurally unconfirmed effector proteins with matches to structural homologues obtained with the DALI server. The DALI Z-score and RMSD of the predicted model with respect to the homologue match is included.

Effector Name	Fungal Species	Structural Homologue (PDB ID)	DALI Z-Score	RMSD (Å)
Avr2	*Fulvia fulva*	Outer surface protein A/B (1P4P)	4.5	2.8
AvrStb6	*Zymoseptoria tritici*	Type II secretion system protein J (5BW0)	4.6	2.2
BAS2	*Magnaporthe oryzae*	Serine–threonine protein kinase PLK4 (5LHX)	3.2	4.0
MiSSP7	*Laccaria bicolor*	Deneddylase (4TT0)	3.5	2.1
BAS4	*Magnaporthe orzae*	Rod-shape determining protein MREB (4CZM)	5.8	3.0
PstSCR1	*Puccinia striiformis* f. sp. *tritici*	Type VI secretion system protein (5H7Z)	5.6	3.5
ECP1	*Fulvia fulva*	Mediator of RNA polymerase II transcription subunit (5N9J)	6.0	2.9
AVRA1	*Blumeria graminis* f. sp. *hordie*	DUF1795 domain containing protein (1MIL)	5.9	3.4
Iug6	*Magnaporthe oryzae*	Fermitin family homologue 1 (4BBK)	5.0	2.8
		Neurobeachin (1MI1)	5.4	2.1
PstPEC6	*Puccinia striiformis* f. sp. *tritici*	Protein of unknown function (DUF1696) with pleckstrin (3DCX)	3.0	2.4
		UDP-N-acetylmuramoylalanine-D-glutamyl-lysine-D-A (2AM1)	3.6	3.2

**Table 5 ijms-24-06262-t005:** List of structurally unconfirmed effector proteins and their structural classification (CATH).

Effector Name (Species)	CATH Classification (PDB ID Match)	C: Class A: Architecture T: Topology H: Homologous Superfamily	SSAP Score, RMSD (Å)	Ref.
Avr2 *(Fulvia fulva)*	2.30.30.370 (1WZO)	Mainly beta. Roll. SH3 type barrels. FAH Domain	78.5, 4.3	[41]
AvrStb6 *(Zymoseptoria tritici)*	3.10.450.210 (2P0X)	Alpha beta. Roll. Nuclear Transport Factor 2, Chain A.	75.2, 3.9	[42]
BAS2 *(Magnaporthe oryzae)*	3.30.390.90 (3D9X)	Alpha beta. Two-layer sandwich. Enolase-like domain 1.	76.4, 7.2	[43]
MiSSP7 *(Laccaria bicolor)*	1.10.8.550 (2L09)	Mainly alpha. Orthogonal bundle. Helicase, Ruva protein, domain 3. Proto-chlorophyllide reductase 57 kD subunitB.	72.3, 2.7	[44]
BAS4 *(Magnaporthe oryzae)*	3.30.429.10 (3EJ7)	Alpha beta. Two-layer sandwich. Macrophage migration inhibitory factor (T and H).	80.1, 5.9	[43]
PstSCR1 *(Puccinia striiformis* f. sp. *tritici)*	3.10.450.210 (2P0X)	Alpha beta. Roll. Nuclear transport factor 2, chain A.	80.5, 3.3	[45]
ECP1 *(Fulvia fulva)*	2.60.40.1180 (1LWJ)	Mainly beta. Sandwich. Immunoglubulin-like. Golgi alpha-mannoside 2	76.2, 3.3	[46]
AVRA1 *(Blumeria graminis* f. sp. *hordei)*	3.30.160.190 (2JYA)	Alpha beta. Two-layer sandwich. Double stranded RNA binding domain. Atu1810-like domain.	74.8, 4.0	[47]
Iug6 *(Magnaporthe oryzae)*	3.30.160.60 (4F6M)	Alpha beta. Two-layer sandwich. Double stranded RNA binding domain. Classic zinc finger.	78.9, 2.3	[48]
PstPEC6 *(Puccinia striiformis* f. sp. *tritici)*	2.20.25.10 (2JS4)	Mainly beta. Single sheet. N-terminal domain of TfIIb.	74.6, 3.9	[49]

## Data Availability

The data supporting the findings of this study are available within the article and its Appendix A. The authors would be happy to provide any additional data upon request.

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
