# Peer review of "Ab Initio Modelling of the Structure of ToxA-like and MAX Fungal Effector Proteins"

_ijms, 2023, doi:10.3390/ijms24076262_

Round 1

Reviewer 1 Report

The Manuscript [ijms-2217391-v1] entitled (Ab initio modelling of the structure of ToxA-like and MAX 2fungal effector proteins) used Rosetta ab initio modelling to predict the structures of members of the ToxA-like and MAX effector families for which experimental structures are known to validate this method. Rosetta was found to successfully predict the structure of fungal effectors in the ToxA-like and MAX families, as well as phenotypically-validated but structurally-unconfirmed effector sequences. This manuscript is well designed with good explanation and has good data those are introduced and written very well. BUT, some comments for the authors that are considered as minor comments:

Comments

1-     Lines 127-134: these should be in discussion.

2-     line 247: should be in discussion.

3-     Lines 319-323: [In general, …… bad one], should be in discussion

4-     Lines 337, 351, 360, 372, 472, 473: these information with the references should be in discussion

5-     Lines 409-414: [The main reason ….. unviable], should be in discussion.

6-     Lines 492-493: these should be in section of Methods

7-     Lines 530-531: [These effectors … [73–82]]. should be in discussion.

8-     Lines 561-562: [T2SS is ….. environment [83]]. should be in discussion.

9-     Lines 574-577, 621, 633: [T6SS is known ….. intoxication [84]]. should be in discussion.

10- Lines 636-640: [ProSA estimates a ….like [56]. should be in discussion.

11-  Conclusion: Should be summarized to be not more 10 lines for the important findings and recommendations

Author Response

Point 1: Lines 127-134: these should be in discussion.

Response 1: Sentence has been shifted to the Discussion section (now Line 667).

Point 2: Line 247: should be in discussion.

Response 2: Sentence was retained since it is necessary to discuss the energy landscape before presenting the corresponding results (now Line 261).

Point 3: Lines 319-323: [In general, …… bad one], should be in discussion

Response 3: Sentence has been shifted to the Discussion section (now Line 700).

Point 4: Lines 337, 351, 360, 372, 472, 473: these information with the references should be in discussion

Response 4: References related to this information were deleted since they are mostly related to the PDB IDs, which can be found in the Supplementary Table S1 and in the Introduction section. (Line 351, 365, 374 and 386). References related to analysis tools/databases were also deleted since they had already been cited in the Methods section (now Line 486 and 487).

Point 5: Lines 409-414: [The main reason ….. unviable], should be in discussion.

Response 5: Sentences were retained since they explain the reasoning of finding the best model within the largest cluster (now Line 423-428).

Point 6: Lines 492-493: these should be in section of Methods

Response 6: Sentence has been deleted since it is in fact redundant because of a similar sentence in the Methods section (Line 227).

Point 7: Lines 530-531: [These effectors … [73–82]]. should be in discussion.

Response 7: Sentence has been shifted to the Discussion section (now Line 836).

Point 8: Lines 561-562: [T2SS is ….. environment [83]]. Should be in discussion.

Response 8: Sentence has been shifted to the Discussion section (now Line 856).

Point 9: Lines 574-577, 621, 633: [T6SS is known …..intoxication [84]]. should be in discussion.

Response 9: Sentence in Line 574 has been shifted to the Discussion section (now Line 857). Sentences in Lines 621 and 633 have been shifted to the Methods section (now Lines 202 and 207, respectively).

Point 10: Lines 636-640: [ProSA estimates a ….like [56]. Should be in discussion.

Response 10: Sentence has been shifted to the Methods section (now Line 203).

Point 11: Conclusion: Should be summarized to be not more 10 lines for the important findings and recommendations

Response 11: The Conclusion section has been shortened and suitable sentences from the Conclusion have been shifted to the Discussion section (Line 769). The Conclusion section, however, continues to provide not only a summary of findings but also a brief discussion of future directions of this research.

Reviewer 2 Report

Rozzano et al. describe in detail the ab initio modeling of the structures of a set of fungal proteins for which there is evidence that the corresponding genes are involved in the infection of plants by the fungi. These proteins are about 250 residues long, secreted by the fungi, and are collectively known as “effectors”.

The authors test the computational folding tool Rosetta on 9 effectors for which experimentally determined three-dimensional structures are available. Then they model the structures of 38 effectors, and 10 of these are described in high detail.

The authors say that accurate structural models for three of the effectors with known structures, and less accurate models for the other six, were achieved. They also state that successful predictions were made on the other effectors, which are likely correct representations of their structures, and that they should be validated experimentally. They propose that these models open up the possibility of predicting effector-plant partner interactions.

The authors explain that they use the TM-score as a measure of accuracy because it is based on global topology and depends on little on local structure variations. It seems to me that global topology is a reasonable criterion for assessing global fold recognition or perhaps belonging to a given structural family. But accuracy should be evaluated based on global and local structures. TM-scores larger than 0.5 are considered to indicate that two structures have the same fold, but a prediction with a TM-score just above 0.5 is not an accurate model. It seems to me that only one effector structure is predicted with high accuracy (AvsPib) because the C-alpha RMSD is very low (0.7 A). Naturally, its TM-score is high (0.92). The predictions for the other 8 effectors are of much smaller accuracy.

If only one structure out of nine effectors is accurately predicted, the confidence in the predictions for the other effectors cannot be high. Perhaps structural similarities can be inferred, as the authors do, but predicting effectors-plant partner interactions cannot, or at least the models do not provide a significant advance for this prediction. The results of the work are, therefore, of limited relevance.

Predicting the structures of proteins with many irregular structures (loops) and with disordered regions is very hard, especially if there are few sequence homologs. The authors mention “… the importance of identifying unique structural features for each effector protein, which might require customisation of Rosetta ab initio modelling to obtain more accurate models.” One unique feature might be disulfide bridges. The authors say that the effectors are rich in cysteine residues (although I see only one in AvrL5667, and none in AvrPib). But perhaps the effectors without known structures have many cysteines and their possible pairing in disulfide bridges can be used to select plausible structural models.

The authors could improve the clarity and consistency of the text and figures, for example (not an exhaustive list):

-       Use the same color code in figures showing results on different proteins, as does not occur in figures 5 and 6,

-       The expression “structurally-unconfirmed effector sequences” is confusing, a structure does not confirm the nature of a sequence as being an effector,

-       Make better descriptions in figure legends and tables: table S3 does not show sequences (table S4 does), and figure S1 does not show superimpositions,

-       Show the ribbon diagrams of the same protein always in the same orientation, unless necessary for a specific reason,

-       The names of the proteins or genes should be the same all along the paper (PST_Pec6 or PstPec6, but not one or the other in different places),

-       Some gene or protein effector names are not found in the PHI-base (for example, PSTPec6)

-       The names of the fungi in the supplementary tables do not always match the names of the fungi in the PHI-base.

Author Response

Point 1: The authors explain that they use the TM-score as a measure of accuracy because it is based on global topology and depends on little on local structure variations. It seems to me that global topology is a reasonable criterion for assessing global fold recognition or perhaps belonging to a given structural family. But accuracy should be evaluated based on global and local structures. TM-scores larger than 0.5 are considered to indicate that two structures have the same fold, but a prediction with a TM-score just above 0.5 is not an accurate model. It seems to me that only one effector structure is predicted with high accuracy (AvsPib) because the C-alpha RMSD is very low (0.7 A). Naturally, its TM-score is high (0.92). The predictions for the other 8 effectors are of much smaller accuracy.

Response 1: TM-score was applied to quantify similarity based on global topology and to assess the global fold to try to determine what family the effector model belongs, to rather than to aim to predict a perfect effector model. The sentence regarding TM-score as a measure of accuracy (Line 696) has been changed to instead indicate similarity. We state in the Results section that TM-scores above 0.5 (which indicate structural similarity) can be used reliably to discriminate a good enough model from the pool of models generated by Rosetta, which we termed ‘best model’.

The use of TM-scores larger than 0.5 has been reported in the literature to quantify structural similarity, such as in the study cited in Reference #50 (Xu and Zhang, 2010). In addition, a recently published work (Reference #81 by Seong and Krasileva, 2023) involving the modelling of effectors corroborated the use of TM-score >0.5 to select similar effector structures. This study was published around the same time of submission of this study, and it is now cited in the Discussion section. Consequently, there is good evidence that TM-score is an adequate indicator that can be applied in the study of fungal effector structures, whereas relying on RMSD has been shown to be inaccurate, as explained in the first part of the Discussion section. We have included suggestions and potential improvements that could be applied in the future to obtain better modelling or accuracy of the models. The high accuracy in the prediction of AvrPib was achieved due to the absence of long disordered loops within the structure, considering that the presence of loops is a common feature of fungal effector structures, as explained in the Discussion section.

Point 2: If only one structure out of nine effectors is accurately predicted, the confidence in the predictions for the other effectors cannot be high. Perhaps structural similarities can be inferred, as the authors do, but predicting effectors-plant partner interactions cannot, or at least the models do not provide a significant advance for this prediction. The results of the work are, therefore, of limited relevance.

Response 2: The statement related to predicting effector-plant partner interactions has been removed (Line 875). The findings in this study involving the methodology and suggestions for improvement constitute the first in implementing a structure-based approach using non-deep learning ab initio modelling for effector identification. Effector identification is a significant challenge in sequence-based effector prediction (Jones et al, 2018 and Jones et al, 2021), since they are a highly mutable, sequence diverse group of loosely functionally-related proteins. Despite perhaps incremental improvements relative to other structural prediction studies, the findings of this research are important in the field of plant pathology as a new method to overcome past limitations in effector prediction.

Point 3: Predicting the structures of proteins with many irregular structures (loops) and with disordered regions is very hard, especially if there are few sequence homologs. The authors mention “… the importance of identifying unique structural features for each effector protein, which might require customisation of Rosetta ab initio modelling to obtain more accurate models.” One unique feature might be disulfide bridges. The authors say that the effectors are rich in cysteine residues (although I see only one in AvrL5667, and none in AvrPib). But perhaps the effectors without known structures have many cysteines and their possible pairing in disulfide bridges can be used to select plausible structural models.

Response 3: An initial part of the research in this study had indeed included cysteine pairing during modelling. However, determining the possible pairing of cysteine/disulphide bridges adds another challenge since there are no existing tools that could provide accurate predictions of cysteine pairing when applied to available effector template structures. Consequently, in this paper we decided to exclude any parameters involving disulphide bridges since they would negatively affect the predicted structural models (without known structures) and inaccurate cysteine pairing could detrimentally decrease the quality and reliability of the predicted models.

In addition, other non-effector secreted proteins of fungal pathogens also contain disulphide bridges, e.g. hydrophobins, so this is not a defining characteristic of effectors. Secretion, length/MW and cysteine content are certainly useful filters to narrow down effector candidates, and which have been intensively studied (for example Jones et al 2018, Bertazonni et al 2021). However this still leaves ~200-500 candidates in a typical fungal proteome, thus necessitating better ways of reducing the size of this set to only the highest probable effector candidates.

Point 4: The authors could improve the clarity and consistency of the text and figures, for example (not an exhaustive list):

Use the same color code in figures showing results on different proteins, as does not occur in figures 5 and 6.

Response 4: We have decided to retain the original colour scheme of these figures since the purpose of using a different colour scheme in Figures 5 and 6 (gold-cyan/ gold-blue) is to differentiate effector models belonging to different effector families (ToxA and MAX). The same applies to Figures 7 and 8, with the aim of representing models generated by different modelling tools for different effector families.

Point 5: The expression “structurally-unconfirmed effector sequences” is confusing, a structure does not confirm the nature of a sequence as being an effector,

Response 5: Effector sequences that were used in this study are all phenotypically validated effectors with no structures available.

Point 6: Make better descriptions in figure legends and tables: table S3 does not show sequences (table S4 does), and figure S1 does not show superimpositions,

Response 6: We have amended the captions for several figures and tables in the manuscript as well as in the Supplementary Information, including Tables S3 and Table S4 and Figure S1.

Point 7: Show the ribbon diagrams of the same protein always in the same orientation, unless necessary for a specific reason,

Response 7: Ribbon diagrams for Figures 5-8 were oriented in the same orientation. Ribbon diagrams in Figure 9 could not be oriented similarly since they are effectors from different structural families and indicators on the orientation is indicated by the rainbow colour scheme, from dark blue at the N-terminus to dark red at the C-terminus.

Point 8: The names of the proteins or genes should be the same all along the paper (PST_Pec6 or PstPec6, but not one or the other in different places),

Response 8: We have synchronised the names of all proteins and genes used in this study, including PstPEC6 throughout the manuscript and in the Supplementary Information.

Point 9: Some gene or protein effector names are not found in the PHI-base (for example, PSTPec6)

Response 9: PHI-base is not a complete record of all known fungal effectors, such as in the case of PstPEC6. The sequences used in this study were obtained based from the training dataset used in EffectorP (Sperschneider et al, 2018), which were based on PHI-base and also contain effectors obtained from the literature. Information on the effector sequences used in this study can be found in UniProt database, and UniProt identifiers for each effectors are provided in the Supplementary Table S3.

Point 10: The names of the fungi in the supplementary tables do not always match the names of the fungi in the PHI-base.

Response 10: Fungi have alternate/updated names for the same organism, such as Passalora fulva is synonymous with Fulvia fulva or Cladosporium fulvum. These name changes/alternate names are common in fungal biology.

Round 2

Reviewer 2 Report

Rozzano et al. have revised their manuscript with minor changes. I think that the changes, improve the presentation of the results. I also think that the authors could make it better. They could avoid sentences like the following one:

We demonstrate in the benchmarking studies that ab initio modelling can predict the structure of effectors from both the ToxA-like and MAX families with sufficient accuracy to open up new pathways for effector discovery.

Because no such demonstration is presented in their article.

They could also clarify/modify the following:

-       ¿Are their models of the ToxA-like and MAX effectors consistent with the disulfide bridges present in the crystal structures, based on distances between the two cysteine residues?

-       The nomenclature of the proteins could be more consistent, in the text, tables, and figures (for instance Avr2 in line 732, Tables 2 and 3, and in figure 9, but AVR2 in Table 4).

-       A fungi species may have two or three correct equivalent names, but it would be more consistent to use the same names as in the PHI database.

Author Response

Response to Reviewer 2 Comments

Point 1: They could avoid sentences like the following one: “We demonstrate in the benchmarking studies that ab initio modelling can predict the structure of effectors from both the ToxA-like and MAX families with sufficient accuracy to open up new pathways for effector discovery. “ Because no such demonstration is presented in their article.

Response 1: The sentence has been rephrased accordingly, including removing the phrase related to “new pathways for effector discovery”. (Line 652)

Point 2: They could also clarify/modify the following: Are their models of the ToxA-like and MAX effectors consistent with the disulfide bridges present in the crystal structures, based on distances between the two cysteine residues?

Response 2: We have clarified in the manuscript that the modelling of the structures of effector templates reported did not involve the use of experimentally-confirmed disulfide bond constraints, ultimately because it would not be possible to apply such constraints to the modelling of effector candidates (Line 174). The predicted models of ToxA-like and MAX effector templates that contain disulfide bridges were, as expected, not fully consistent in these regions with the crystal structures. This is mainly due to the models lacking secondary structure predicted in the regions involved in disulfide bridge formation, which affects the overall conformation in that region. Most of the disulfide bridges in the crystal structures of the effectors are formed between a cysteine located at the start of a beta-strand and a cysteine located in a disordered loop. Due to the conformational flexibility of the disordered region, this resulted in deviations between the pairing cysteines. This was observed in effector templates that contain disulphide bridges: Avr2 (Fol), ToxB, AvrPik, Avr1CO39, AvrPia and AvrPiz-t. The distances between the paired cysteines in the predicted models were predominantly 5-10 Ang, potentially allowing additional modelling to be undertaken to re-introduce the disulfide bridges. This has been clarified in the manuscript (Line 359).

Point 3: The nomenclature of the proteins could be more consistent, in the text, tables, and figures (for instance Avr2 in line 732, Tables 2 and 3, and in figure 9, but AVR2 in Table 4).

Response 3: The nomenclature has been standardised throughout the main document and in the Supplementary Materials, particularly for Avr2. In this study, two different Avr2 effectors were considered, which we have named as Avr2 (Fol) and Avr2 (F. fulva), corresponding to the Avr2 effector templates of Fusarium oxysporum f. sp. lycopersici and the Avr2 effector candidate of Fulvia fulva.

Point 4: A fungi species may have two or three correct equivalent names, but it would be more consistent to use the same names as in the PHI database.

Response 4: Similar names of fungi species have been updated in the main document and in the Supplementary Materials as per the PHI-base. This includes updating Passalora fulva to Fulvia fulva.

Round 3

Reviewer 2 Report

The new information about the consistency of the models with the available crystal structures regarding the disulfide bridges will be useful to help the readers evaluate the quality of the models. However, there is a misconception in the discussion of the possible reasons for this inconsistency. The authors wrongly identify disordered regions with ordered regions without a regular secondary structure. Regular secondary structures (helices, sheets, turns) are structures with “canonical” dihedral angles and H-bond patterns, but regions with non-regular structures are also ordered, just not ordered into regular secondary structures. Disordered regions do not show up in the crystal structures. A disordered region (with very high conformation flexibility) does not give rise to electron density. Therefore, it is not appropriate to discuss the discrepancy between the models and the crystal structures in terms of disorder. The distances between the cysteine residues in the models, too long to be involved in disulfide bridges, is not a consequence of bad predictions for disordered regions, it is a consequence of bad predictions for the ordered regions that do not form regular secondary structures.

Author Response

Response to Reviewer 2 Comments

Point 1: The new information about the consistency of the models with the available crystal structures regarding the disulfide bridges will be useful to help the readers evaluate the quality of the models. However, there is a misconception in the discussion of the possible reasons for this inconsistency. The authors wrongly identify disordered regions with ordered regions without a regular secondary structure. Regular secondary structures (helices, sheets, turns) are structures with “canonical” dihedral angles and H-bond patterns, but regions with non-regular structures are also ordered, just not ordered into regular secondary structures. Disordered regions do not show up in the crystal structures. A disordered region (with very high conformation flexibility) does not give rise to electron density. Therefore, it is not appropriate to discuss the discrepancy between the models and the crystal structures in terms of disorder. The distances between the cysteine residues in the models, too long to be involved in disulfide bridges, is not a consequence of bad predictions for disordered regions, it is a consequence of bad predictions for the ordered regions that do not form regular secondary structures.

Response 1: We agree with the reviewer about our choice of words. We have modified our discussion to avoid referring to 'disordered regions" and instead, in this case, we simply refer to loop regions in the proteins of interest.
